

# Foreshock cavitons and spontaneous hot flow anomalies: A statistical study with a global hybrid-Vlasov simulation

Vertti Tarvus[1], Lucile Turc[1], Markus Battarbee[1], Jonas Suni[1], Xóchitl Blanco-Cano[2], Urs Ganse[1], Yann Pfau-Kempf[1], Markku Alho[1], Maxime Dubart[1], Maxime Grandin[1], Andreas Johlander[1], Konstantinos Papadakis[1], and Minna Palmroth[1,3]

[1]Department of Physics, University of Helsinki, Helsinki, Finland
[2]Instituto de Geofísica, Universidad Nacional Autónoma de México, Mexico City, Mexico
[3]Finnish Meteorological Institute, Helsinki, Finland

**Correspondence:** Vertti Tarvus (vertti.tarvus@helsinki.fi)

**Abstract.** The foreshock located upstream of Earth's bow shock hosts a wide variety of phenomena related to the reflection of solar wind particles from the bow shock and the subsequent formation of ultra-low frequency (ULF) waves. In this work, we investigate foreshock cavitons, which are transient structures resulting from the non-linear evolution of ULF waves, and spontaneous hot flow anomalies (SHFAs), which evolve from cavitons as they accumulate suprathermal ions while being

carried to the bow shock by the solar wind. Using the global hybrid-Vlasov simulation model Vlasiator, we have conducted a statistical study in which we track the motion of individual cavitons and SHFAs in order to examine their properties and evolution. In our simulation run where the interplanetary magnetic field (IMF) is directed at a sunward-southward angle of $45°$, continuous formation of cavitons is found up to $\sim 11$ Earth radii ($R_E$) from the bow shock (along the IMF direction), and caviton-to-SHFA evolution takes place within $\sim 2 \, R_E$ from the shock. A third of the cavitons in our run evolve into SHFAs,

and we find a comparable amount of SHFAs forming independently near the bow shock. We compare the properties of cavitons and SHFAs to prior spacecraft observations and simulations, finding good agreement. We also investigate the variation of the properties as a function of position in the foreshock, showing that transients close to the bow shock are associated with larger depletions in the plasma density and magnetic field magnitude, along with larger increases in the plasma temperature and the level of bulk flow deflection. Our measurements of the propagation velocities of cavitons and SHFAs agree with earlier studies,

showing that the transients propagate sunward in the solar wind rest frame. We show that SHFAs have a greater solar wind rest frame propagation speed than cavitons, which is related to an increase in the magnetosonic speed near the bow shock.

## 1 Introduction

As the supermagnetosonic solar wind interacts with Earth's magnetosphere, a curved bow shock forms upstream of Earth. The

bow shock slows the solar wind down to submagnetosonic speeds while compressing and heating the flow before it is deflected





by the magnetopause separating the solar wind and the magnetosphere. At the part of the bow shock where the interplanetary magnetic field (IMF) connects to the shock, the solar wind electrons and ions that are reflected from the shock are able to propagate far back into the upstream. The backstreaming particles form a foreshock region upstream of the quasi-parallel shock (where the shock normal and the IMF have an angle $\theta_{\mathrm{Bn}} < 45°$), contrasting the more abrupt shock crossing found at
the quasi-perpendicular shock ($\theta_{\mathrm{Bn}} > 45°$).

The foreshock is characterised by instabilities arising due to the interaction between the backstreaming particles and the incoming solar wind. Depending on the angle $\theta_{\mathrm{Bn}}$, a range of interesting features are found in Earth's foreshock. These include different shock-reflected suprathermal ion populations (e.g., Fuselier, 1995; Kempf et al., 2015), different types of ultra-low frequency (ULF) waves (e.g., Hoppe et al., 1981; Greenstadt et al., 1995; Wilson, 2016) and a variety of transient structures
(See Eastwood et al. 2005 for a review of the foreshock, and Zhang and Zong 2020 for a recent review of foreshock transients).

Foreshock cavitons are transients that arise in Earth's foreshock due to the non-linear evolution of ULF waves. Cavitons are found amidst ULF waves and suprathermal ions, and they are identified as localised, highly correlated depressions in plasma density and magnetic field magnitude, bounded by rims where these parameters are enhanced. The extents of cavitons are of the order of 1 Earth radius ($R_{\mathrm{E}}$) with the plasma density and magnetic field magnitude decreasing by $\sim 50\%$ from their ambient
solar wind values on average (Blanco-Cano et al., 2009, 2011; Kajdič et al., 2011, 2013). The temperature inside cavitons is similar to their surroundings, and they do not contain significant bulk flow deflections (Kajdič et al., 2013). While cavitons propagate sunward in the solar wind rest frame, they are carried back to the bow shock by the solar wind (Kajdič et al., 2011). Hybrid simulations (kinetic ions and fluid electrons) and spacecraft observations have demonstrated that the solar wind rest frame propagation speed often exceeds the Alfvén speed (Blanco-Cano et al., 2011; Wang et al., 2020).

Initially, the existence of cavitons was predicted using hybrid simulations (Lin, 2003; Lin and Wang, 2005). Also using hybrid simulations, Omidi (2007) and Blanco-Cano et al. (2009) proposed that under a radial IMF configuration, cavitons form when two types of ULF waves, parallel-propagating right- or left-hand polarised waves and obliquely propagating linearly-polarised compressive waves, steepen and interact with each other. The presence of cavitons in spacecraft data was later confirmed with Cluster observations, where they were found to exist for a variety of different solar wind and IMF conditions
(Kajdič et al., 2011; Blanco-Cano et al., 2011; Kajdič et al., 2013).

The term "caviton" was introduced by Blanco-Cano et al. (2009) to distinguish the transient type from foreshock cavities, which are similarly characterised by simultaneous density and magnetic depressions surrounded by rims of enhanced density and magnetic field magnitude (e.g., Sibeck et al., 2002; Schwartz et al., 2006; Billingham et al., 2011). Foreshock cavities form at isolated IMF flux tubes that are connected to the bow shock. As the field lines in these flux tubes are filled up with
shock-reflected suprathermal particles, the plasma on the field lines expands, leading to a cavity where the plasma density and magnetic field magnitude are depressed (Schwartz et al., 2006). Unlike cavitons, foreshock cavities only contain suprathermal particles in their interiors but not in their surroundings due to their formation mechanism. Another difference is that cavitons are always surrounded by ULF waves, which is not necessarily true for foreshock cavities (e.g., Kajdič et al., 2017).

Another class of transients found in Earth's foreshock are hot flow anomalies (HFAs) (e.g., Schwartz et al., 1985; Schwartz,
1995; Lucek et al., 2004) which form when a tangential discontinuity in the IMF interacts with the bow shock and confines





suprathermal ions in its vicinity. HFAs are also characterised by simultaneous depressions in plasma density and magnetic field magnitude bounded by enhancements, but they further display a significantly elevated temperature and a highly deviated bulk flow inside them. Signatures of HFAs in the absence of IMF discontinuities led to the identification of a separate transient phenomenon, spontaneous hot flow anomalies (SHFAs) (Zhang et al., 2013; Omidi et al., 2013). SHFAs evolve from cavitons

when they are advected to the bow shock by the solar wind and they accumulate suprathermal ions. Global hybrid simulations have shown that the arrival of multiple SHFAs at the bow shock can cause the bow shock surface to erode (Blanco-Cano et al., 2018), and that SHFAs can impact the magnetosheath between the bow shock and the magnetopause by causing the formation of magnetosheath filamentary structures (Omidi et al., 2014a) or magnetosheath cavities and magnetosheath jets (Omidi et al., 2016). Omidi et al. (2016) also reported that the pressure variations caused by SHFAs and magnetosheath cavities can lead to

the motion of the magnetopause.

Cavitons and SHFAs have been studied statistically in the past by using observations from the Cluster spacecraft. Kajdič et al. (2013) studied a set of 92 cavitons observed between 2001-2006, and Kajdič et al. (2017) later extended the statistical study to include other transients as well, including 19 observations of SHFAs from 2003-2011. Both cavitons and SHFAs were found for various solar wind conditions, and they were observed within 9 $R_E$ and 6 $R_E$ from the bow shock, respectively

(as measured along the IMF direction). Cavitons were found preferentially during stronger IMF, lower solar wind density and larger solar wind and Alfvén speeds. Based on the speed of the solar wind and the duration of the cavitons in spacecraft data, Kajdič et al. (2013) estimated the extents of the cavitons to be in the range of $1 - 13$ $R_E$. Inside cavitons, the plasma density and magnetic field magnitude decreased by $20\% - 90\%$ from their solar wind values, with an average decrease of $50\%$. The observed SHFAs were very depleted, showing average decreases of $90\%$ in the density and magnetic field magnitude from

their solar wind values.

Using global hybrid simulations, Omidi et al. (2014b) studied the variation of SHFAs' properties with parameters such as Alfvén Mach number ($M_A$), IMF cone angle and the transients' distance from the Sun-Earth line. A low SHFA formation rate was found for $M_A \leq 3$, with the rate increasing with $M_A$. The formation rate showed no clear dependence on the IMF cone angle, and a run with a cone angle of $90°$ demonstrated that SHFAs can form even under such geometry. Omidi et al.

(2014b) found that increasing $M_A$ resulted in the SHFAs having stronger features, such as larger magnetic enhancements at the transients' edges along with greater ion temperatures and greater levels of solar wind deceleration. Further, the duration of the SHFAs did not show dependence on the Mach number, which suggests that their size also increases with $M_A$. Formation of SHFAs occurred preferentially close to the bow shock nose, where the transients exhibited larger temperatures compared to the ones located further away from the nose. Close to the bow shock nose, SHFAs also tended to have stronger magnetic

enhancements and greater levels of solar wind deceleration, but these features were found to be less noticeable at lower $M_A$.

The statistical studies of Kajdič et al. (2013, 2017) and Omidi et al. (2014b) have helped in establishing a global picture of cavitons and SHFAs in Earth's foreshock, but there are still many details in this picture that are not well understood. In particular, open questions concerning the evolution of the transients include: Where and how often do cavitons emerge in the foreshock? How do their properties evolve as they are advected towards the bow shock by the solar wind? Do all cavitons

evolve into SHFAs? In the present paper, we aim to understand the evolution of cavitons and SHFAs in Earth's foreshock





by performing a statistical study of these transients with the global hybrid-Vlasov simulation model Vlasiator (von Alfthan et al., 2014; Palmroth et al., 2018). By tracking the motion of individual cavitons and SHFAs in a simulation run with an IMF vector directed at a $45°$ sunward-southward angle, we have studied the properties and the evolution of 1445 transients, the largest statistical sample up to date. We first examine and compare the physical properties of cavitons and SHFAs, such as their density and magnetic field depressions, ion temperature, bulk flow speed and size. We then present statistical maps of the formation locations of cavitons and SHFAs, and study the rate of caviton-to-SHFA evolution throughout the foreshock. We also investigate the variation of the properties of the transients as a function of the distance from the bow shock and the Sun-Earth line. Finally, we calculate the propagation velocities of the transients, presenting the first large sample statistical results of the propagation of cavitons and SHFAs.

## 2 Methods

### 2.1 Vlasiator simulation

We study foreshock cavitons and SHFAs using the Vlasiator model (von Alfthan et al., 2014; Palmroth et al., 2018), which is a hybrid-Vlasov code designed for performing global simulations of Earth's magnetosphere and the space surrounding it, capable of resolving ion kinetic effects. In the hybrid-Vlasov approach, ions are modelled using velocity distribution functions (VDFs), which evolve according to Vlasov's equation, and electrons are modelled as a cold, massless charge-neutralising fluid. In Vlasiator, Vlasov's equation is coupled to Maxwell's equations and the system is closed with Ohm's law including the Hall term. Ion VDFs are discretised on a 3D Cartesian grid where a sparse grid representation is utilised in order to maintain numerical efficiency.

The simulation run used in our study is the same that has been previously used by Blanco-Cano et al. (2018) to study cavitons and SHFAs, and by Hoilijoki et al. (2019) to study dayside magnetic reconnection and flux transfer events (FTEs). The run is 2D in ordinary space and 3D in velocity space. The simulation is performed in the noon-midnight meridional plane and it uses the Geocentric Solar Ecliptic (GSE) coordinate system. In this system, Earth is located at the origin, the X-axis is directed towards the Sun, the Z-axis is perpendicular to the ecliptic plane and the dusk-directed Y-axis completes the coordinate system. Our simulation domain covers $[-48.66\,\mathrm{R_E}, 64.35\,\mathrm{R_E}]$ in the X-direction, $[-59.65\,\mathrm{R_E}, 39.24\,\mathrm{R_E}]$ in the Z-direction and a single cell in the Y-direction. The simulation has a spatial resolution of $300$ km and a velocity space resolution of $30$ $\mathrm{km\,s^{-1}}$. The solar wind enters the simulation domain from the positive X-boundary, and other boundaries of the domain use copy conditions. In the out-of-plane Y-direction, periodic boundary conditions are used. The inner boundary of the simulation corresponds to a perfectly conducting ionosphere represented by a static Maxwellian distribution, located at $30\,000$ km ($\sim 4.7$ $\mathrm{R_E}$) from the origin. Earth's geomagnetic field is modelled as a 2D line dipole (Daldorff et al., 2014) with the tilt neglected.

In our run, the solar wind has a velocity of $\mathbf{v}_{\mathrm{SW}} = (-750, 0, 0)$ $\mathrm{km\,s^{-1}}$. Protons are the only ion species present, with a number density $n_{\mathrm{SW}} = 1.0$ $\mathrm{cm^{-3}}$ and temperature $T_{\mathrm{SW}} = 0.5$ MK in the solar wind, corresponding to fast solar wind conditions. The IMF is set to $\mathbf{B}_{\mathrm{SW}} = (3.54, 0, -3.54)$ nT, directed sunward and southward with a cone angle of $45°$ and a total magnitude of 5 nT. With these parameters, the solar wind in our simulation run has $\beta = 0.7$, magnetosonic Mach number





$M_{\mathrm{ms}} = 5.6$ and Alfvén Mach number $M_{\mathrm{A}} = 6.9$. The proton inertial length $c\omega_{\mathrm{pi}}^{-1}$ and the proton gyroradius $r_{\mathrm{i}}$ in the solar

wind are $c\omega_{\mathrm{pi}}^{-1} = 228$ km and $r_{\mathrm{i}} = 214$ km, respectively. The duration of the simulation is $1437.5$ s, with the state available at $0.5\ s$ intervals.

## 2.2    Detecting and tracking foreshock transients

To detect cavitons and SHFAs, we define them as structures where both the proton number density and the magnetic field magnitude are below $80\%$ of their respective ambient solar wind values. These criteria have been previously utilised by Blanco-

Cano et al. (2018) in their Vlasiator study and by Kajdič et al. (2013, 2017) in Cluster observations. In our simulation run, these thresholds are $0.8\ \mathrm{cm}^{-3}$ for the density and $4.0$ nT for the magnetic field magnitude. In order to distinguish SHFAs from cavitons, we require that SHFAs fulfill an additional criterion of having $\beta > 10$ in at least $60\%$ of the cells belonging to a transient. This criterion was also previously used by Blanco-Cano et al. (2018), and a value of 10 is chosen as it is significantly above the typical foreshock $\beta$.

To study cavitons and SHFAs evolving in time, we identify individual transients and track their motion over their lifetime. We begin the tracking procedure at $t = 900.0$ s, when the bow shock and the foreshock appear well formed, and continue the tracking until $t = 1437.5$ s, resulting in a total time interval of $537.5$ s. We first identify all simulation cells fulfilling the caviton criteria at each timestep of the tracking time interval. We exclude the cells located earthward of the bow shock by introducing a boundary based on a 4th order polynomial fit of the shock. The boundary is updated at each timestep in order to account for

the outward motion of the bow shock due to 2D effects, and it is shifted earthward by $3.5\ \mathrm{R_E}$ from the calculated position of the fit to account for irregularities in the bow shock. After we have found all the cells that fulfill the caviton criteria, we sort the cells into individual transients.

Figure 1 shows two zoomed-in views of the simulation run used in this study at $t = 1012.5$ s. A global view of the foreshock is shown in Figure 1a, where cavitons and SHFAs are highlighted in black (no distinction between the transient types). A

dashed line shows the boundary delimiting the area earthward of the bow shock. Cavitons and SHFAs are found deep in the foreshock, as is suggested by the fluctuations seen in the background colourmap displaying the proton number density. Several transients exhibit elongated shapes and are found together in "chains" that are aligned with the direction of the IMF. Figure 1b shows a further zoomed-in view that illustrates the interior structure of cavitons and SHFAs. The colour scale inside the transients represents the "depth" of the transients, which we define as the maximum value between the proton number density

and the magnetic field magnitude normalised by their solar wind values. Many of the transients have complex interiors, where either one or multiple density/magnetic field minima can be observed. We find that such complex structures arise as multiple transients interact with each other and merge. Conversely, large transients occasionally split up and form multiple smaller transients.

We track the motion of cavitons and SHFAs by identifying overlapping transients at consecutive timesteps. The overlapping

transients are either advecting, splitting into multiple transients or merging together. Using this method, we reconstruct the history of each tracked transient. We require the tracked transients to have a size of at least 5 cells ($\sim 0.011\ \mathrm{R_E^2}$) and set a lower limit for the overlap between transients at $25\%$ of the smaller transient's cell count. Disappearing transients are monitored



for overlapping for 5 seconds (10 timesteps), which allows us to consistently track shallow and momentarily disappearing transients. After the tracking, we categorise the transients using the SHFA criterion. We use three general categories, which are "pure cavitons" that never fulfill the SHFA criterion during their lifetime, "pure SHFAs" that always fulfill the SHFA criterion during their lifetime and "evolving transients" that fulfill the SHFA criterion only part of their lifetime.

## 3 Results

During our tracking time interval spanning 537.5 s, we have tracked a total of 1445 unique transients. We detected 1272 (88.0%) independently forming transients and 133 (9.2%) transients that split off from existing transients. In total, we detected 274 merging events between the tracked transients. At the beginning of our tracking interval, we detected 40 (2.8%) transients, which we have omitted from our formation and propagation statistics as it is unclear whether these transients existed prior to the tracking interval. At any given time, we observe $29 - 72$ simultaneous transients in the foreshock, and find the average number of simultaneous transients to be 51. At all times, the number of observed cavitons exceeds the number of observed SHFAs. The percentage of cavitons varies between $56.8 - 88.6\%$, with an average percentage of $74.8\%$. In terms of transient evolution, we find that pure cavitons make up half of the tracked transients (725, $50.2\%$), while pure SHFAs (342, $23.7\%$) and evolving transients (378, $26.1\%$) represent roughly a quarter each. In the latter category we find mainly cavitons that evolve into SHFAs (341, $23.6\%$), while the rest (37, $2.5\%$) are either SHFAs that turn into cavitons or transients that change their classification momentarily.

### 3.1 Physical properties

In Table 1, we present statistics of the physical properties of the tracked cavitons and SHFAs. We have looked at the extrema of the properties occurring during the transients' lifetime, such as the minimum density, magnetic field magnitude and bulk flow speed and the maximum temperature and area. For each property, we have categorised the transients into cavitons and SHFAs based on their type at the time when the extremum of the property was reached. In Figure 2, we visualise some of the distributions of the properties presented in Table 1.

Inside cavitons, the total proton number density $n$ and magnetic field magnitude $|\boldsymbol{B}|$ decrease from their ambient solar wind values by $20 - 61\%$ and $20 - 88\%$, respectively. Inside SHFAs, the decreases in $n$ range between $20 - 73\%$ and the decreases in $|\boldsymbol{B}|$ between $20 - 94\%$. We remind here that the lower limit is due to the criteria we use to detect the transients. On average, both $n$ and $|\boldsymbol{B}|$ decrease by $28\%$ inside cavitons, while the average values for SHFAs are $36\%$ for $n$ and $39\%$ for $|\boldsymbol{B}|$. Figures 2a and 2b show that the decreases inside cavitons are mostly concentrated near the transient detection limit while the decreases inside SHFAs are distributed more evenly across their respective value ranges. We see that the number of SHFA datapoints exceeds the number of caviton datapoints when $n$ and $|\boldsymbol{B}|$ decrease below $\sim 65\%$ of their solar wind values, indicating that SHFAs tend to be more depleted than cavitons.

As expected, we observe a significant number of suprathermal protons inside SHFAs, while cavitons contain only small amounts of them. We consider the suprathermal population to contain all protons outside of the solar wind core population,





which is centered around the solar wind bulk flow velocity within a $500\ \mathrm{km\,s^{-1}}$ radius sphere in velocity space. The suprather-mal proton number density $n_{\mathrm{st}}$ inside SHFAs reaches up to $78\%$ of the solar wind proton number density and is at $21\%$ on average, while the average is only $4\%$ inside cavitons. Figure 2c shows that $n_{\mathrm{st}}$ rarely exceeds $15\%$ of the solar wind density inside cavitons, but routinely reaches high values inside SHFAs. To verify that the observed high values of $n_{\mathrm{st}}$ inside SHFAs are not due to high values inside individual cells but represent an overall increase of the amount of suprathermal protons, we

repeated the same analysis using the maximum average value of $n_{\mathrm{st}}$ instead of the overall maximum, and found the results to be similar. While the suprathermal proton density inside SHFAs increases, the transients also experience a further depletion of the solar wind core population. We find that the solar wind core proton number density $n_{\mathrm{c}}$ decreases up to $99\%$ from the ambient solar wind proton number density inside SHFAs, with an average decrease of $49\%$. The largest depletions of the solar wind core population inside SHFAs are found very close to the shock (see Section 3.3 later), which may have occured as a

result of the core population being deflected outside the $500\ \mathrm{km\,s^{-1}}$ sphere in the velocity space used for defining the solar wind core population.

In accordance with the high number of suprathermal protons, SHFAs exhibit higher proton temperatures and larger reduc-tions in the bulk flow speed compared to cavitons. Notably, in Figure 2d we see a clear cut-off temperature in the distribution of the proton temperatures separating the caviton and SHFA distributions at $\sim 7\ \mathrm{MK}$ (14 times the solar wind ion temperature

$T_{\mathrm{SW}}$). The cut-off is likely due to our $\beta$-based criterion that was used to categorise the transients. On average, the maximum proton temperature inside SHFAs reaches $12.3\ \mathrm{MK}$ ($24.6\,T_{\mathrm{SW}}$), and it generally does not exceed $30\ \mathrm{MK}$ ($60\,T_{\mathrm{SW}}$). We found only few instances of higher temperatures up to $\sim 50\ \mathrm{MK}$ ($100\,T_{\mathrm{SW}}$) alongside very high $\beta$ ($\sim 1000$) when individual transient cells interacted with the bow shock. Figure 2e shows that the bulk flow inside SHFAs appears significantly deviated from the solar wind flow, as expected from the combination of the weakened solar wind core population and the enhanced backstreaming

suprathermal population found inside SHFAs. The average value of the bulk flow speed $|\boldsymbol{v}_{\mathrm{flow}}|$ inside SHFAs is $604.5\ \mathrm{km\,s^{-1}}$, which corresponds to a $19.4\%$ decrease from the solar wind bulk flow speed of $750\ \mathrm{km\,s^{-1}}$. The level of deceleration varies, with some SHFAs even having $|\boldsymbol{v}_{\mathrm{flow}}|$ well below $400\ \mathrm{km\,s^{-1}}$ ($\gtrsim 47\%$ decrease). Cavitons generally contain only minor bulk flow deflections, showing an average $|\boldsymbol{v}_{\mathrm{flow}}|$ of $704.1\ \mathrm{km\,s^{-1}}$ ($6.1\%$ reduction) and only few examples of $|\boldsymbol{v}_{\mathrm{flow}}|$ below $600$ $\mathrm{km\,s^{-1}}$. We stress that the perceived decreases in the bulk flow speed are likely not due to the deceleration of the solar wind

core but instead due to the suprathermals' effect on the calculation of the bulk flow speed from the velocity distribution function (See Parks et al. 2013 for demonstration of this effect inside HFAs in spacecraft observations).

Cavitons and SHFAs do not significantly differ in terms of transient size. Figure 2f shows that the maximum areas of the two transient types are distributed similarly. The transients are mostly small, with roughly $78\%$ of all tracked transients having maximum areas less than $0.2\ \mathrm{R_E}^2$. Cavitons have a slightly larger average maximum area ($0.16\ \mathrm{R_E}^2$) than SHFAs ($0.12\ \mathrm{R_E}^2$),

which could be due to SHFAs forming only near the bow shock, where they do not have time to grow large. We find that the largest transients (with areas of up to $1.91\ \mathrm{R_E}^2$) have grown as a result of several transients merging together, while the largest naturally growing (no mergers involved) transient had an area of $1.02\ \mathrm{R_E}^2$.

We looked for possible correlations between the physical properties presented in Table 1. In Figure 3, we present a set of three scatterplots illustrating the correlation between selected properties that represent the characteristics of cavitons and





SHFAs. The values are chosen such that the values on the X-axis represent extrema over transient lifetime and the values on the Y-axis are extrema taken from the same timestep as the values on the X-axis. We have calculated the Pearson correlation coefficient in each scatterplot for all transients, cavitons and SHFAs separately.

In Figure 3a, a good correlation is found between the minima of the total number density and magnetic field magnitude inside cavitons, showing a correlation coefficient $C = 0.79$. We find weaker correlation inside SHFAs ($C = 0.57$) which is in part due to the large amount of suprathermal protons, leading to smaller decreases in total number density compared to the magnetic field magnitude. By replacing the total number density with the solar wind core number density (not shown), we find the correlation between the density and magnetic field magnitude improving inside SHFAs ($C = 0.69$) while it remains unaffected inside cavitons. This suggests that the proton number density and magnetic field magnitude decrease in concert inside cavitons, but the evolution becomes more complex when cavitons accumulate suprathermals and turn into SHFAs.

Figure 3b shows that there is a clear correspondence between the maximum suprathermal number density and the maximum temperature inside cavitons ($C = 0.79$) when the suprathermal number density is low ($\lesssim 30\%$ of $n_{\mathrm{SW}}$). As the suprathermal number density increases (up to $\sim 78\%$ of $n_{\mathrm{SW}}$), we see a widening range of temperatures and the relationship between the suprathermal density and the temperature becomes non-linear. In Figure 3c, the minima of the bulk flow speed steadily show lower values when the maxima of the suprathermal number density increase, which supports the notion that the observed slowdown is an effect of suprathermals on the calculation of the bulk flow speed instead of actual deceleration of the solar wind. While cavitons contain only few suprathermals, the bulk flow speed inside cavitons appears to decrease down to $\sim 600 - 650$ $\mathrm{km\,s^{-1}}$ due to the presence of suprathermals, which suggests that even a low amount of suprathermals can have a notable impact on the observed bulk flow speed.

## 3.2 Transient formation and evolution

Throughout our tracking interval, we find a steady rate of transient formation with no extended periods of increased/decreased formation. By taking a 10 second moving average, we find that the transient formation rate fluctuates between $\sim 1 - 4$ transients forming per second, with an average of $\sim 2.4$. In Figure 4, we present the cumulative counts of forming cavitons, forming SHFAs and cavitons turning into SHFAs as a function of the distance from the bow shock measured along the IMF direction. For this measurement, we consider the bow shock to be located at the first cell in which the solar wind core temperature is at least 4 times the ambient solar wind temperature (see Wilson et al. (2014) and Battarbee et al. (2020) for the application of this method on spacecraft and simulation data, respectively). Since some transients change their classification between a caviton and an SHFA multiple times, we consider only the final change from a caviton into an SHFA counting as a caviton turning into an SHFA.

Figure 4a shows that 99% of all caviton formation occurs within $\sim 11~\mathrm{R_E}$ from the bow shock. We have excluded 6 outliers from the plot, which were detected to form at $12.6 - 39.5~\mathrm{R_E}$. The formation of cavitons is most abundant within $\sim 4~\mathrm{R_E}$ from the shock, where $\sim 70\%$ of the caviton formation takes place. However, we do not find cavitons forming very close to the bow shock at distances of $\lesssim 0.3~\mathrm{R_E}$. Figures 4b and 4c demonstrate that SHFAs are observed within $\sim 4~\mathrm{R_E}$ from the bow shock (note the different scale on the X-axis compared to Figure 4a), where they either form independently or evolve from cavitons.





We find that the rates of SHFA formation and caviton-to-SHFA evolution are very similar, but find a slightly higher number

of SHFAs evolving from cavitons ($N = 341$) compared to SHFAs forming independently ($N = 277$). The rate of emerging SHFAs (either forming or evolving from cavitons) is at its highest within $\sim 2\ \mathrm{R_E}$ from the bow shock and declines rapidly at larger distances. When moving towards the shock, the number of SHFAs relative to cavitons increases, and we find that the SHFA formation rate surpasses the caviton formation rate at $\sim 0.8\ \mathrm{R_E}$. Despite this, we still find a notable amount of cavitons near the bow shock and not all cavitons evolve into SHFAs. Within $1.5\ \mathrm{R_E}$ from the shock, a total of 180 cavitons form, of

which $\sim 59\%$ evolve into SHFAs.

In Figure 5, we visualise the locations of forming cavitons (Figures 5a and 5b), forming SHFAs (Figures 5c and 5d) and cavitons turning into SHFAs (Figures 5e and 5f) as 2D heatmaps using two different coordinate systems. In the left column of Figure 5 we show the locations using the regular Cartesian GSE-coordinate system, while in the right column of Figure 5 we show the same locations using a coordinate system in which the X-axis corresponds to the distance from the bow shock

along the IMF direction, and the Y-axis displays the "nose angle". We define the nose angle as the angle from the Sun-Earth line measured from Earth, and use it to quantify the transients' distance from the Sun-Earth line along the bow shock. The latter coordinate system has the advantage of being independent of the movement of the bow shock, and hence we use it to supplement the data shown in GSE-coordinates as the bow shock in our simulation run experiences some outward motion due to 2D effects.

Figures 5a and 5b show that the caviton formation region significantly expands with increasing nose angle. We suspect that the full extent of the region might be cut by the lower boundary of the simulation domain, since we do not see a notable decrease in the formation rate near the boundary. In Figures 5c-5f, we see that the region in which SHFAs form or evolve from cavitons has a constant width of $\sim 2\ \mathrm{R_E}$ without significant variation with the distance from the Sun-Earth line, but find a low rate of formation/evolution at the flank of the bow shock near the lower boundary of the simulation domain. This suggests that

our simulation approximately captures the full extent of the region where SHFAs exist in the foreshock. We find that at nose angles $\gtrsim 80°$ SHFAs more commonly evolve from cavitons as opposed to forming independently.

### 3.3 Variation of physical properties with the location in the foreshock

We have studied the physical properties of the tracked transients for possible dependencies on the transients' location relative to the bow shock. In Figures 6 and 7, we present two series of box-and-whiskers plots showing how the properties listed in

Table 1 vary with the distance from the bow shock (along the IMF direction) and the nose angle (as defined in the previous section). We compare the variation of the properties of cavitons and SHFAs in side-by-side plots. The data shown contains the properties' extrema from each timestep of the tracked transients' lifetimes (as opposed to overall extrema in Table 1), which we categorise into cavitons and SHFAs depending on the type at each timestep.

Figure 6 shows that when cavitons and SHFAs are located close to the bow shock, they generally tend to have stronger

features, such as larger decreases in the proton number density (Figures 6a, 6b, 6e and 6f), magnetic field magnitude (Figures 6c and 6d) and bulk flow speed (Figures 6m and 6n) and larger increases in the proton temperature (Figures 6i and 6j). These features are more pronounced inside SHFAs compared to cavitons. Far from the shock, the density/magnetic depressions inside





cavitons are mostly shallow. The suprathermal proton number density (Figures 6g and 6h) displays a sharp increase at $1\,\mathrm{R_E}$ from the bow shock, which is also manifested inside SHFAs as an increase in the proton temperature and a decrease in the bulk

flow speed. The low amount of suprathermals beyond $1\,\mathrm{R_E}$ suggests that the accumulation of suprathermals occurs principally very close to the bow shock. Figures 6o and 6p demonstrate that the maximum areas of the observed transients increase towards the bow shock, but the low median areas ($\sim 0.1 - 0.2\,\mathrm{R_E}^2$) indicate that transients are generally small at all distances. There appears to be no significant difference between the size distributions of cavitons and SHFAs.

   Figure 7 suggests that there is no single trend controlling the properties of cavitons and SHFAs as the nose angle varies. A

notable feature at the flank of the bow shock at nose angles $\gtrsim 80°$ is the presence of prominent transients in the upper/lower quartiles of the data, especially visible in the magnetic field magnitude, solar wind core/suprathermal proton number density, $\beta$, bulk flow speed and area. Further analysis shows that these transients are a result of several transients merging together. The region where these transients are found coincides with the region of most abundant caviton formation (Figure 5b), which provides suitable conditions for mergers between transients. Using the same simulation run as we have used in this work,

Blanco-Cano et al. (2018) previously reported on the formation of a large magnetosheath cavity at the flank of the bow shock in response to several SHFAs arriving at the shock. Our statistical results further illustrate that for an IMF oriented at a $45°$ cone angle, the flank of the bow shock is a favourable region for the formation of large transients, which can in turn lead to the formation of magnetosheath cavities and erosion of the bow shock. The large transients resulting from mergers can have complex shapes (see Figure 1), which may also explain the irregular shapes of some of the cavitons seen in multi-spacecraft

observations (Kajdič et al., 2011).

   Apart from the large transients present at the flank of the bow shock, the properties of cavitons in the interquartile data range appear uniform, and we do not find evidence of significant variation in cavitons' properties with the nose angle. At the flank of the bow shock, we find some large cavitons with high suprathermal counts which are not classified as SHFAs possibly due to their size, as our SHFA criterion requires $\beta > 10$ in at least $60\%$ of a transient's cells. Such situation could arise from a merger

between a caviton and an SHFA, resulting to the new transient having a high suprathermal count and a low percentage of cells with $\beta > 10$. For the properties of SHFAs, we observe a clear nose angle dependence in the proton temperature, showing increasing values towards the bow shock nose. Due to the ambiguity in the variation of other SHFA properties, we repeated our analysis using only transients that are not involved in mergers (not shown). This way, in addition to the proton temperature and $\beta$ increasing towards the bow shock nose, we found a similar, but less distinctive trend in the bulk flow speed, which shows

decreasing values towards the bow shock nose. We did not find signs of notable nose angle dependence in other properties of SHFAs.

### 3.4   Transient propagation

Our method of tracking cavitons and SHFAs has allowed us to measure the propagation velocities of the transients in a straightforward manner. First, we have calculated the lifetimes of the transients, which range between $0.5 - 217.5$ s with an average

lifetime of $20.1$ s. Then, we have calculated the distances travelled by the transients by following the position of each transient's magnetic field minimum. The average overall distance travelled by a transient is $1.94\,\mathrm{R_E}$, with an average displacement





of $-1.88$ R$_E$ in the X-direction (Earth-Sun direction) and $-0.40$ R$_E$ in the Z-direction (out-of-ecliptic direction). Thus, on average, the motion of the transients in the simulation frame of reference is directed anti-sunward and southward. Furthermore, we find transient motion in the northward and sunward directions to be negligible. Since the solar wind in our simulation run
flows along the X-axis, any out-of-ecliptic motion of the transients is not due to the solar wind motion.

In Figures 8a and 8b, we have plotted the lifetimes of the transients against their over-the-lifetime displacements in the X-direction (Figure 8a) and the Z-direction (Figure 8b). In each direction, we have made linear fits that estimate the propagation velocities of the transients in the simulation frame of reference. To limit the effects of transient interaction and deformation, we have omitted interacting (splitting, merging) transients from Figure 8. We have calculated separate fits for pure cavitons
($N = 446$), pure SHFAs ($N = 202$) and evolving transients ($N = 198$) to highlight differences between different transient types.

In the X-direction, pure cavitons propagate at the greatest velocity, which we estimate to be $\sim -651$ km s$^{-1}$. The X-directional propagation velocity of pure SHFAs is notably slower, at $\sim -546$ km s$^{-1}$, and the transition from cavitons to SHFAs is highlighted by the intermediate propagation velocity of evolving transients ($\sim -613$ km s$^{-1}$). We find excellent
correlation between the lifetime and the displacement in the X-direction for all transient types, showing that the motion of the transients is uniform as they are advected by the solar wind towards the bow shock. Furthermore, we find that all transient types show X-directional propagation velocities that are slower than the solar wind ($-750$ km s$^{-1}$), indicating that all transient types have a sunward-directed velocity component in the solar wind rest frame. Accordingly, the X-directional propagation velocities in the solar wind rest frame are $\sim 99$ km s$^{-1}$ for pure cavitons, $\sim 204$ km s$^{-1}$ for pure SHFAs and $\sim 137$ km s$^{-1}$ for evolving
transients. In the Z-direction, pure cavitons have the slowest (southward directed) propagation velocity at $\sim -101$ km s$^{-1}$ while pure SHFAs have the fastest propagation velocity at $\sim -155$ km s$^{-1}$. Evolving transients again display an intermediate propagation velocity ($\sim -133$ km s$^{-1}$). We note that in the solar wind rest frame, the trends in the Z- and X-directions are similar, with cavitons propagating slower than SHFAs in both directions.

Based on the fits calculated in Figures 8a and 8b, we find a total solar wind rest frame propagation velocity of $\sim 142$
km s$^{-1}$ for pure cavitons, $\sim 256$ km s$^{-1}$ for pure SHFAs and $\sim 191$ km s$^{-1}$ for evolving transients. The solar wind rest frame propagation of pure cavitons is directed at a sunward and southward angle of $\sim 45.4°$, indicating that the propagation approximately follows the IMF direction ($45°$). We find a similar propagation angle for evolving transients ($\sim 44.1°$) while the propagation is somewhat more sunward oriented for pure SHFAs ($\sim 37.3°$).

To understand the notably different propagation velocities of cavitons and SHFAs, we investigated the relationship between
the propagation velocity of the transients and the magnetosonic speed in the foreshock. In Figure 8c, we have plotted the over-lifetime median values of the magnetosonic speeds measured at the centers of the transients (which we take to be the magnetic minima) against the solar wind rest frame propagation speeds of the transients. Based on visual inspection, we conclude that the magnetosonic speed has similar values inside the transients and in their surroundings, and thus we consider the magnetosonic speeds measured inside the transients to be representative of the surrounding medium.
The results presented in Figure 8c suggest that the solar wind rest frame propagation speed of the transients increases in concert with the local magnetosonic speed. We find that there is an increase in the magnetosonic speed near the bow shock,





leading to SHFAs having a larger propagation speed in the solar wind rest frame compared to cavitons. A linear fit in Figure 8c estimates the solar wind rest frame propagation speed of the transients to be $\sim 0.77 v_{\mathrm{ms}}$, with $v_{\mathrm{ms}}$ being the local magnetosonic speed. However, we note that this is a crude estimate, as we have used a temporal median value of the magnetosonic speed and

large variation is present in the propagation speeds of the transients. We find that the poor correlation coefficient ($C = 0.58$) in Figure 8c is in part due to the presence of small, short-lived transients for which the propagation speed may attain very large values ($\gtrsim 400 \, \mathrm{km \, s^{-1}}$). We repeated our analysis using only transients which have a minimum lifetime of $5 \, \mathrm{s}$ (not shown), and obtained a linear fit with a similar slope ($0.77 \pm 0.03$) and an increased correlation coefficient ($C = 0.72$).

## 4   Discussion

Our choice for the criterion which we use to separate cavitons and SHFAs (SHFAs must have $\beta > 10$ in at least $60\%$ of transient cells) appears to be suitable, since we observe a clear difference in the amount of suprathermal protons inside cavitons and SHFAs, and we are able to reproduce the characteristics of SHFAs established in existing literature (e.g., Zhang et al., 2013; Omidi et al., 2013). True to their name, the SHFAs in our simulation run are associated with high temperatures and high levels of bulk flow deflection due to the large quantities of suprathermals inside them, while cavitons exhibit only minor heating and

flow deflection since they contain very few suprathermals. However, we note that using $\beta$ as an SHFA criterion may lead to transients being classified based on the magnetic field magnitude instead of the effects associated exclusively with suprathermal ions (e.g. increase in temperature and bulk flow deflection), which is most apparent in Figure 7 at the far flank of the bow shock, where large reductions in the magnetic field magnitude inside SHFAs can be seen along with only little heating.

In accordance with Cluster spacecraft observations (Kajdič et al., 2013, 2017), we find the decreases in the proton number

density and magnetic field magnitude inside cavitons to be well correlated. The correlation coefficient between these quantities inside cavitons is similar in spacecraft statistics of Kajdič et al. (2017) ($C = 0.85$) and our study ($C = 0.79$). These decreases have weaker correlation inside SHFAs, which is apparent in both spacecraft statistics ($C = 0.30$) and our results ($C = 0.57$). While the weak correlation in spacecraft observations may be in part due to low statistics ($N = 19$), our results suggest that the weak correlation compared to cavitons is caused by the presence of suprathermals, which can counter the decrease in the

total proton number density and lead to uneven decreases in the number density and magnetic field magnitude.

The Cluster statistics of Kajdič et al. (2017) demonstrated that SHFAs contain larger density and magnetic field magnitude depletions than cavitons, and our results show agreement with this notion. However, the depletions in our simulation run appear shallow compared to spacecraft observations. In our simulation run, the average maximum decrease in the proton number density (magnetic field magnitude) from its ambient solar wind value is $28\%$ ($28\%$) inside cavitons and $36\%$ ($39\%$) inside

SHFAs. In the spacecraft statistics of Kajdič et al. (2017), the average decreases of the density and magnetic field magnitude are $50\%$ inside cavitons and $90\%$ inside SHFAs. The smaller depletions found in our simulation are in part due to our rigorous transient detection method, which is able to pick up even the smallest transients that may not be resolvable from spacecraft data amidst ULF waves. We assessed the impact of the small transients on the calculation of the average density and magnetic field depletions by recalculating the average values using only transients that have a maximum area of at least $0.5 \, \mathrm{R_E}^2$ over lifetime





($N = 75$). In this manner, we found notable increases in the average depletions, with the average maximum decrease for the proton number density (magnetic field magnitude) being $44\%$ ($45\%$) for cavitons and $55\%$ ($62\%$) for SHFAs. The depths of the transients in our simulation run may also be affected by our method of measuring the depths directly from the input solar wind parameters, which do not fully represent the varying conditions in the foreshock. This could possibly lead to the depths of the transients being underestimated.

Similarly, the sizes of the transients in our simulation appear small compared to those found in spacecraft observations. Using Cluster data, Kajdič et al. (2013) estimated the average caviton extent to be $\sim 4.6$ $\mathrm{R_E}$ for a set of 92 observations. While we have directly measured the areas of the transients in our simulation as opposed to calculating the transient extent in one spatial dimension like in the spacecraft data, the average maximum and overall maximum areas we found ($0.16$ $\mathrm{R_E}^2$ and $1.91$ $\mathrm{R_E}^2$, respectively) indicate a clear size difference. This discrepancy is likely not due to different solar wind conditions in

our simulation and the spacecraft observations. According to Omidi et al. (2014b), the size of the transients should increase with Alfvén Mach number $M_A$. In our simulation, we have $M_A = 6.9$, which is similar to the average Alfvén Mach number during the caviton observations of Kajdič et al. (2013) ($\langle M_A \rangle = 6.5$, with $M_A$ ranging between $2 < M_A < 15$). Instead, we identify two effects that may contribute to the difference in the sizes of the transients. First, the sizes could be affected by our solar wind-based transient detection criteria in the same manner as the decreases in density and magnetic field, leading to the

transients' sizes to be underestimated. Second, Kajdič et al. (2013) have included the rims of enhanced density/magnetic field magnitude associated with cavitons in the calculated extents. This is not done in our study, since our caviton detection criteria only account for the depressions inside the transients. In the first Vlasiator study of cavitons and SHFAs, Blanco-Cano et al. (2018) noted that these rims are not as prominent in Vlasiator as in spacecraft data. This is believed to be an effect of the spatial resolution of the simulation, which can limit the steepening of ULF waves (Pfau-Kempf et al., 2018), and could also affect the

growth of cavitons and SHFAs.

    We find overall good agreement between our study and the spacecraft statistics of Kajdič et al. (2017) regarding the locations of cavitons and SHFAs relative to the bow shock. Like Kajdič et al. (2017), we have measured the transients' distances from the bow shock along the IMF direction. In the spacecraft statistics, cavitons and SHFAs have been observed within $9$ $\mathrm{R_E}$ and $6$ $\mathrm{R_E}$ from the bow shock, respectively. In our simulation run, $99\%$ of all caviton formation takes place within $11$ $\mathrm{R_E}$ from

the bow shock, while all SHFAs are found within $4$ $\mathrm{R_E}$ from the shock. It is noteworthy that the Cluster spacecraft used in the observations of Kajdič et al. (2017) are on an orbit that keeps the spacecraft near the bow shock, which may lead to fewer observations further away from the shock. We also note that the spacecraft statistics span a long time period with varying solar wind/IMF conditions while our single simulation run has a constant solar wind and IMF. It is possible that the distance measurement is affected by the orientation of the IMF, and that the extents of the caviton/SHFA regions change with the solar

wind/IMF conditions. Investigating the influence of parameters such as Alfvén Mach number and IMF cone angle on the extent of the caviton/SHFA region is a possible topic for a future study.

    Our results indicate that the accumulation of suprathermals inside cavitons/SHFAs is closely tied to the transients' distance from the bow shock. This is exemplified by the fact that we see a simultaneous onset of copious SHFA formation and caviton-to-SHFA evolution at $\sim 2$ $\mathrm{R_E}$ from the bow shock (as measured along the IMF direction), occurring at roughly the same





distance at all parts of the quasi-parallel bow shock. Most notably, we observe a significant jump in the suprathermal proton number density, temperature and the level of bulk flow deflection at $\lesssim 1\,\mathrm{R_E}$ from the shock inside both cavitons and SHFAs. We also see overall larger decreases of the proton number density and magnetic field magnitude near the bow shock, which is in agreement with prior numerical studies of cavitons (Blanco-Cano et al., 2009, 2011).

   The transients' distance from the bow shock nose/Sun-Earth line appears to have an impact on the formation and the prop-
erties of SHFAs in our simulation. We find our results to be in good agreement with the findings of Omidi et al. (2014b), who studied the dependence of SHFAs' properties on the distance from the Sun-Earth line using global hybrid simulations with various Alfvén Mach numbers and IMF orientations. Both our study and the study of Omidi et al. (2014b) find that the amount of SHFAs decreases towards the flank of the bow shock. In our simulation run, the decreasing amount of SHFAs at the bow shock flank is contrasted by the abundant formation of cavitons there, which emphasises that there is no decline in the
transient formation rate at the flank but the transients there generally undergo lesser heating/evolution compared to the subsolar region. Similar to Omidi et al. (2014b), we clearly observe that the temperature inside SHFAs increases towards the bow shock nose. The temperature inside the SHFAs in our simulation run ($M_\mathrm{A} = 6.9$) increases to $\sim 8 - 60\,T_\mathrm{SW}$, which is in good agreement with the values obtained by Omidi et al. (2014b) for different Alfvén Mach numbers ($\sim 2 - 20\,T_\mathrm{SW}$ for $M_\mathrm{A} = 5$ and $\sim 5 - 160\,T_\mathrm{SW}$ for $M_\mathrm{A} = 11$). Omidi et al. (2014b) also noted a clear dependence on the distance from the Sun-Earth line
in the bulk flow speed inside SHFAs that favours smaller speeds near the shock nose when the Alfvén Mach number is large ($M_\mathrm{A} = 11$). We also find some evidence of larger reductions in the bulk flow speed inside SHFAs near the bow shock nose, although we do not see as clear variation with the distance from the Sun-Earth line as in the temperature. This could be due to the smaller Alfvén Mach number in our simulation run.

   In agreement with previous studies of cavitons, the transients in our simulation exhibit sunward motion in the solar wind
rest frame while they propagate anti-sunward in the simulation frame. Our estimates for the solar wind rest frame propagation speeds of the transients ($\sim 142 - 256\,\mathrm{km\,s^{-1}}$) are comparable to those found in multi-spacecraft measurements. Kajdič et al. (2011) used a simple timing method to calculate the sunward propagation speeds of two cavitons ($120\,\mathrm{km\,s^{-1}}$, $188\,\mathrm{km\,s^{-1}}$), while Wang et al. (2020) used a variety of multipoint analysis methods to calculate the velocities of 12 cavitons ($\lesssim 300\,\mathrm{km\,s^{-1}}$). Our results suggest that SHFAs propagate at a faster solar wind rest frame speed than cavitons due to an increase in the mag-
netosonic speed near the bow shock. We attribute this increase in the magnetosonic speed to an enhanced pressure/temperature near the shock, leading to an increased sound speed. On the other hand, the Alfvén speed does not show significant dependence on the bow shock distance, but instead fluctuates with the wave activity in the foreshock. Due to this, estimating the Alfvén speed in the foreshock is difficult. With respect to the ambient solar wind, the solar wind rest frame motion of the transients in our simulation run is super-Alfvénic ($1.3 v_\mathrm{A,SW}$ for pure cavitons, $2.4 v_\mathrm{A,SW}$ for pure SHFAs), agreeing with earlier reports of
super-Alfvénic motion of cavitons (Blanco-Cano et al., 2011; Wang et al., 2020).

   In accordance with the results of Wang et al. (2020), our results indicate that cavitons propagate in the direction of the IMF. However, in order to reach definite conclusions on the propagation direction, multiple simulation runs with different IMF orientations should be used. We find that compared to cavitons, the propagation of SHFAs is somewhat more sunward oriented (for an IMF cone angle of $45°$). This may be due to the conditions near the bow shock, where the propagation of SHFAs





is influenced by the highly perturbed flow/magnetic field. Accumulation of ions that have been specularly reflected from the bow shock could also affect the propagation of SHFAs by introducing a velocity component parallel to the local shock normal direction, which near the nose of the bow shock could explain the more sunward oriented propagation of SHFAs.

## 5   Conclusions

In this work, we have statistically studied cavitons and SHFAs in Earth's foreshock using a global hybrid-Vlasov simulation run
conducted in the noon-midnight meridional plane. Our simulation corresponds to fast solar wind conditions, having an Alfvén Mach number $M_\mathrm{A} = 6.9$ and an IMF cone angle of $45°$. We have found that under such conditions, cavitons and SHFAs are a common occurrence in the foreshock, with $\sim$1-4 new transients forming every second and 29-72 transients populating the foreshock in the noon-midnight meridional plane at any given time. We find cavitons to be more numerous than SHFAs, with cavitons making up $\sim 75\%$ of the simultaneously observed transients on average.

The properties of cavitons and SHFAs in our simulation are in good overall agreement with the findings of prior numerical and observational studies. We find the depressions of the plasma density and magnetic field magnitude to be well-correlated inside cavitons, while the correlation is reduced inside SHFAs due to their copious accumulation of suprathermal protons. On average, the depressions are larger inside SHFAs. Clear correspondence is found between the amount of suprathermals and the level of heating and bulk flow speed reduction inside SHFAs. The latter appears to arise due to the suprathermal population's
contribution on the calculation of the bulk flow velocity instead of it being actual slowdown of the solar wind flow. The only clear discrepancy regarding the properties of cavitons/SHFAs between our simulation and spacecraft observations is seen in the transients' sizes and the depths of their density/magnetic depressions, which are smaller in our simulation than in observations. This may be due to the differing methods used in measuring the extents/depths of the transients or the resolution of our simulation being a limiting factor for the growth of the transients.

We find continuous formation of cavitons extending up to $\sim 11$ $\mathrm{R_E}$ from the bow shock (as measured along the IMF direction). The caviton formation region attains its largest extent deep in the foreshock, showing gradual expansion from the bow shock nose towards the flank of the shock. The formation most likely extends beyond the boundary of our simulation domain, which prevents us from determining the full extent of the caviton formation region. Cavitons are principally found to evolve into SHFAs within $\sim 2$ $\mathrm{R_E}$ from the bow shock, where SHFAs are also commonly found forming independently.
$\sim 55\%$ of the observed SHFAs evolve from cavitons, while the rest form independently. Not all cavitons in our run undergo evolution, with only roughly a third of all cavitons turning into SHFAs. SHFAs emerge at similar distances in all parts of the foreshock, although the rate of SHFA formation/caviton evolution declines towards the flank of the bow shock.

The properties of cavitons and SHFAs show some dependence on the transients' position relative to the bow shock. We have found that there are overall increases in the depth of the density/magnetic depressions and the level of heating and
bulk flow speed reduction inside transients as their distance to the bow shock decreases. The number density of suprathermal protons inside the transients is generally low until $\lesssim 1$ $\mathrm{R_E}$ from the shock, showing that the accumulation of suprathermals is concentrated near the bow shock. The SHFAs' distance from the Sun-Earth line has a clear effect on the temperature inside





them, with higher temperatures found near the bow shock nose. There is a similar, but weaker dependency for the bulk flow
speed inside SHFAs, with lower speeds near the nose. The distance from the Sun-Earth line does not appear to have any direct

effect on the properties of cavitons, but our results suggest that mergers between transients can lead to the formation of large
transients deep in the foreshock where the formation of cavitons is most abundant.

The transients in our simulation run propagate sunward in the solar wind rest frame of reference, but they are advected
anti-sunward in the simulation frame of reference by the solar wind. The propagation also has an out-of-ecliptic component,
which is directed southward in our simulation run where the IMF also has a southward component. We have found that the

propagation of cavitons in the solar wind rest frame is aligned with the direction of the IMF, while SHFAs exhibit slightly
more sunward oriented propagation. We find that SHFAs have a greater solar wind rest frame speed than cavitons, which is
related to an increase in the magnetosonic speed near the bow shock. We find an approximate solar wind rest frame transient
propagation speed of $\sim 0.77 v_{\mathrm{ms}}$, with $v_{\mathrm{ms}}$ being the local magnetosonic speed. The solar wind rest frame propagation speed of
the transients exceeds the Alfvén speed $v_{\mathrm{A,SW}}$ in the ambient solar wind, with a speed of $1.3 v_{\mathrm{A,SW}}$ for cavitons and $2.4 v_{\mathrm{A,SW}}$

for SHFAs.

Our study has presented a comprehensive picture of cavitons and SHFAs in Earth's foreshock and has demonstrated the
usefulness of global hybrid-Vlasov simulations in complementing spacecraft observations of foreshock phenomena. Topics
especially suited for global simulations include the propagation and the evolution of foreshock transients, which in spacecraft
studies typically require the usage of multi-spacecraft missions. In future numerical studies, multiple simulation runs with

different solar wind conditions could be used to study the effect of parameters such as the Alfvén Mach number and the IMF
cone angle on the formation, propagation and evolution of cavitons and SHFAs.

*Code availability.* Vlasiator (http://www.helsinki.fi/en/researchgroups/vlasiator/, Palmroth, 2020) is distributed under the GPL-2 open source
license at http://github.com/fmihpc/vlasiator/ (Palmroth and the Vlasiator team, 2020). Vlasiator uses a data structure developed in-house
(https://github.com/fmihpc/vlsv/, Sandroos, 2019). The Analysator software (https://github.com/fmihpc/analysator/, Battarbee and the Vlasi-

ator team, 2020) was used to produce the presented figures. The run described here takes several terabytes of disk space and is kept in storage
maintained within the CSC – IT Center for Science. Data presented in this paper can be accessed by following the data policy on the Vlasiator
web site.

*Author contributions.* VT wrote the first draft of the manuscript and performed the data analysis. LT and MB provided guidance in analysing
and interpreting the results. JS developed the tracking methodology which was adapted by VT for this study. XBC led the work that served

as a starting point for this study, and helped with interpreting the results. MP is the PI of the Vlasiator model and MB, UG and YPK ran
the Vlasiator run used in this study. All coauthors (including MA, MD, MG, AJ and KP) participated in the discussion of the results and the
improvement of the manuscript.



*Competing interests.* The authors declare that they have no conflict of interest.

*Acknowledgements.* We acknowledge the European Research Council for Starting grant 200141-QuESpace, with which Vlasiator was de-
veloped, and Consolidator grant 682068-PRESTISSIMO awarded to further develop Vlasiator and use it for scientific investigations. The
work leading to these results has been carried out in the Finnish Centre of Excellence in Research of Sustainable Space (Academy of Finland
grant number 312351). The work of VT and LT is supported by the Academy of Finland (grant numbers 322544 and 328893). The CSC–IT
Center for Science in Finland is acknowledged for the Sisu supercomputer pilot usage and Grand Challenge award leading to the results
presented here.





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





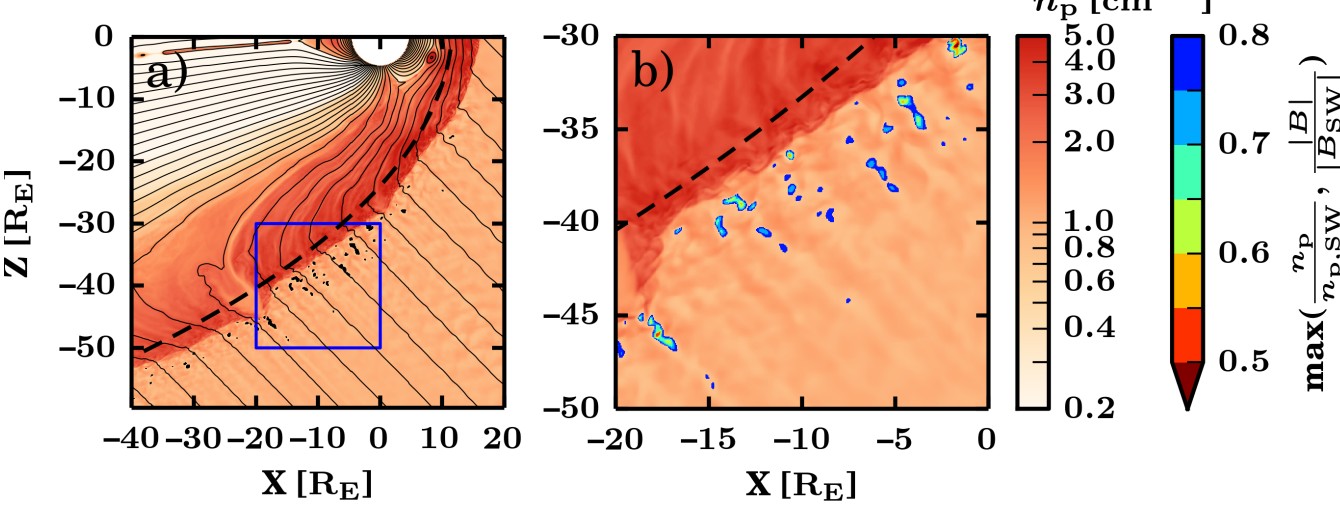

**Figure 1.** Two zoomed-in views of the studied Vlasiator simulation run showing the foreshock at $t = 1012.5$ s. The background colour displays the proton number density $n_p$ (left colourbar) and the thin black continuous lines represent the magnetic field $\boldsymbol{B}$. Panel a) shows cavitons and SHFAs marked in black. The area delimited by the blue square is shown in panel b), where the "depth" (defined as the maximum between $n_p$ and $|\boldsymbol{B}|$ normalised by their solar wind values) of the transients is shown (right colourbar). The thick dashed line represents the boundary behind which cells fulfilling the caviton criteria are ignored.



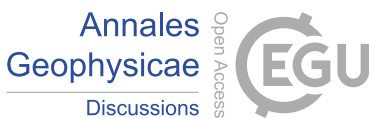

**Table 1.** Statistics of the physical properties of tracked cavitons and SHFAs. The quantities listed in the left column are extrema over each tracked transients' lifetime. For each property, the transients are classified to cavitons and SHFAs depending on their type at the property extremum, with $N$ denoting the number of transients in either class. From top to bottom: Total proton number density $n$, solar wind core proton number density $n_{\mathrm{c}}$, suprathermal proton number density $n_{\mathrm{st}}$, magnetic field magnitude $|\boldsymbol{B}|$, proton temperature $T$, plasma $\beta$, plasma bulk flow speed $|\boldsymbol{v}_{\mathrm{flow}}|$ and transient area $A$. The densities and the magnetic field magnitude are normalised to the constant ambient solar wind values, denoted by "SW".

| Quantity | Cavitons | | | | | SHFAs | | | | |
|---|---|---|---|---|---|---|---|---|---|---|
| | Min. | Max. | Avg. | Med. | N | Min. | Max. | Avg. | Med. | N |
| **Min.** $n/n_{\mathrm{SW}}$ | 0.39 | 0.80 | 0.72 | 0.74 | 864 | 0.27 | 0.80 | 0.64 | 0.66 | 581 |
| **Min.** $n_{\mathrm{c}}/n_{\mathrm{SW}}$ | 0.19 | 0.80 | 0.69 | 0.71 | 816 | 0.01 | 0.76 | 0.51 | 0.55 | 629 |
| **Max.** $n_{\mathrm{st}}/n_{\mathrm{SW}}$ | 2.4e-4 | 0.54 | 0.04 | 0.03 | 747 | 0.03 | 0.78 | 0.21 | 0.17 | 698 |
| **Min.** $|\boldsymbol{B}|/|\boldsymbol{B}_{\mathrm{SW}}|$ | 0.12 | 0.80 | 0.72 | 0.74 | 837 | 0.06 | 0.79 | 0.61 | 0.64 | 608 |
| **Max.** $T$ [MK] | 0.668 | 16.0 | 3.67 | 3.43 | 750 | 4.03 | 50.5 | 12.3 | 10.9 | 695 |
| **Max.** $\beta$ | 1.23 | 543.0 | 8.05 | 6.32 | 752 | 10.2 | 4305.2 | 49.1 | 24.2 | 693 |
| **Min.** $|\boldsymbol{v}_{\mathrm{flow}}|$ [km s$^{-1}$] | 488.2 | 773.7 | 704.1 | 710.2 | 790 | 255.3 | 783.5 | 604.5 | 618.0 | 655 |
| **Max.** $A$ [R$_{\mathrm{E}}{}^2$] | 0.01 | 1.91 | 0.16 | 0.08 | 942 | 0.01 | 1.71 | 0.12 | 0.06 | 503 |



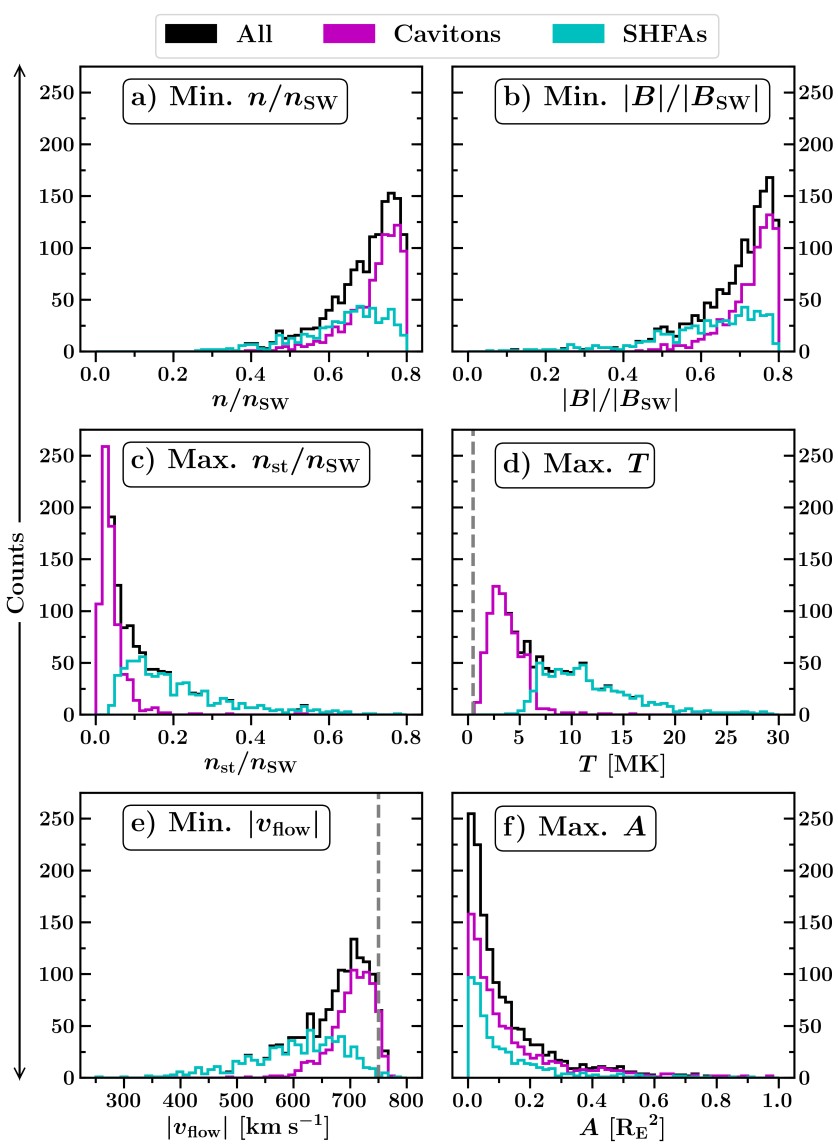

**Figure 2.** Histograms visualising the distributions of selected quantities from Table 1. The grey dashed lines in panels d) and e) represent ambient solar wind values. Panels d) and f) contain outliers beyond 30 MK and 1 $R_E^2$ which have been excluded for clarity.

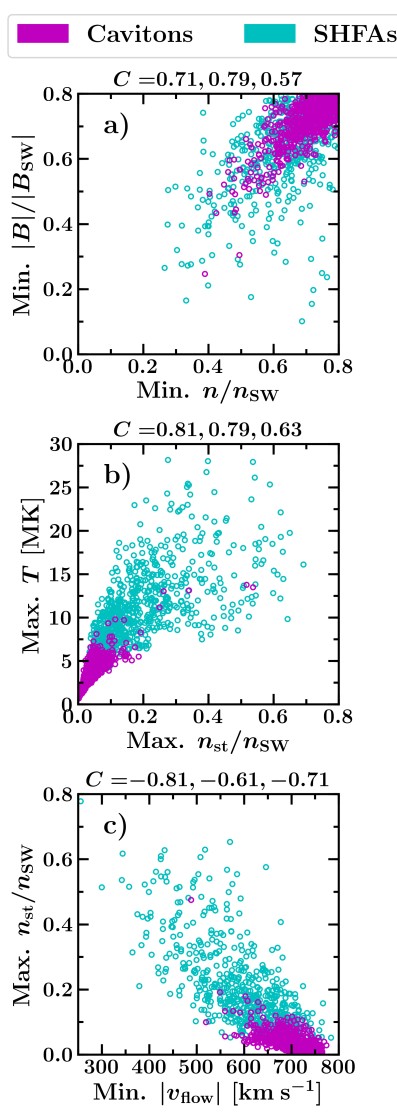

**Figure 3.** Scatterplots showing the correlations between different quantities for the tracked transients. The Pearson correlation coefficients (labelled $C$) are shown above each panel for all transients, cavitons (magenta) and SHFAs (cyan), respectively. The values on the X-axis are overall extreme values over tracked transients' lifetimes and the values on the Y-axis are extreme values taken from the same timestep as the values on the X-axis.



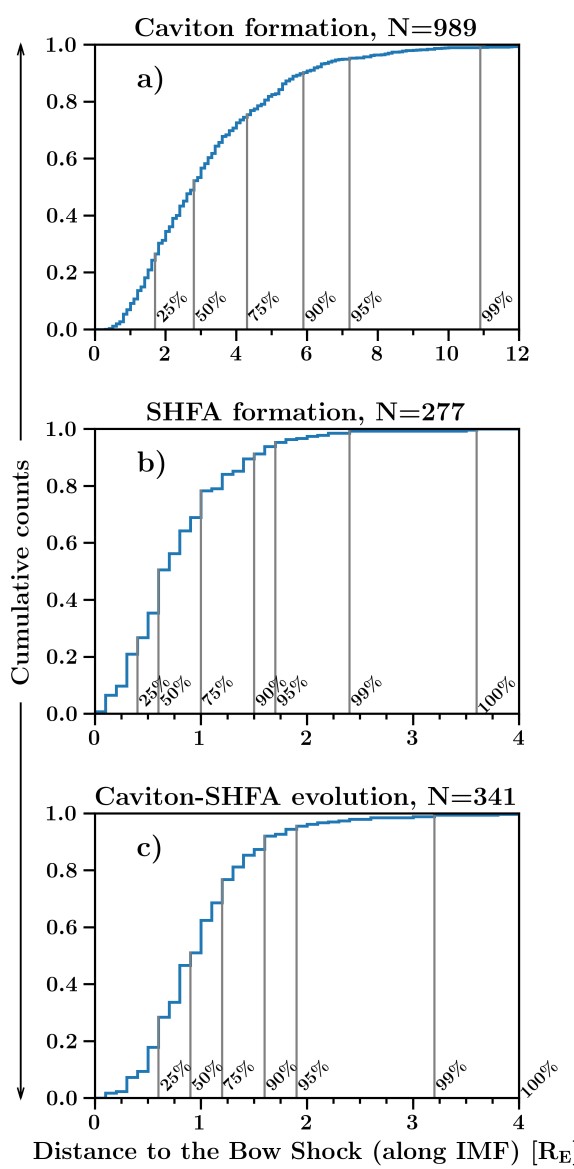

**Figure 4.** Cumulative distributions of a) caviton formation locations, b) SHFA formation locations and c) the locations where caviton-to-SHFA evolution takes place as a function of the distance to the bow shock along the IMF direction (at a $45°$ angle from the Sun-Earth line).



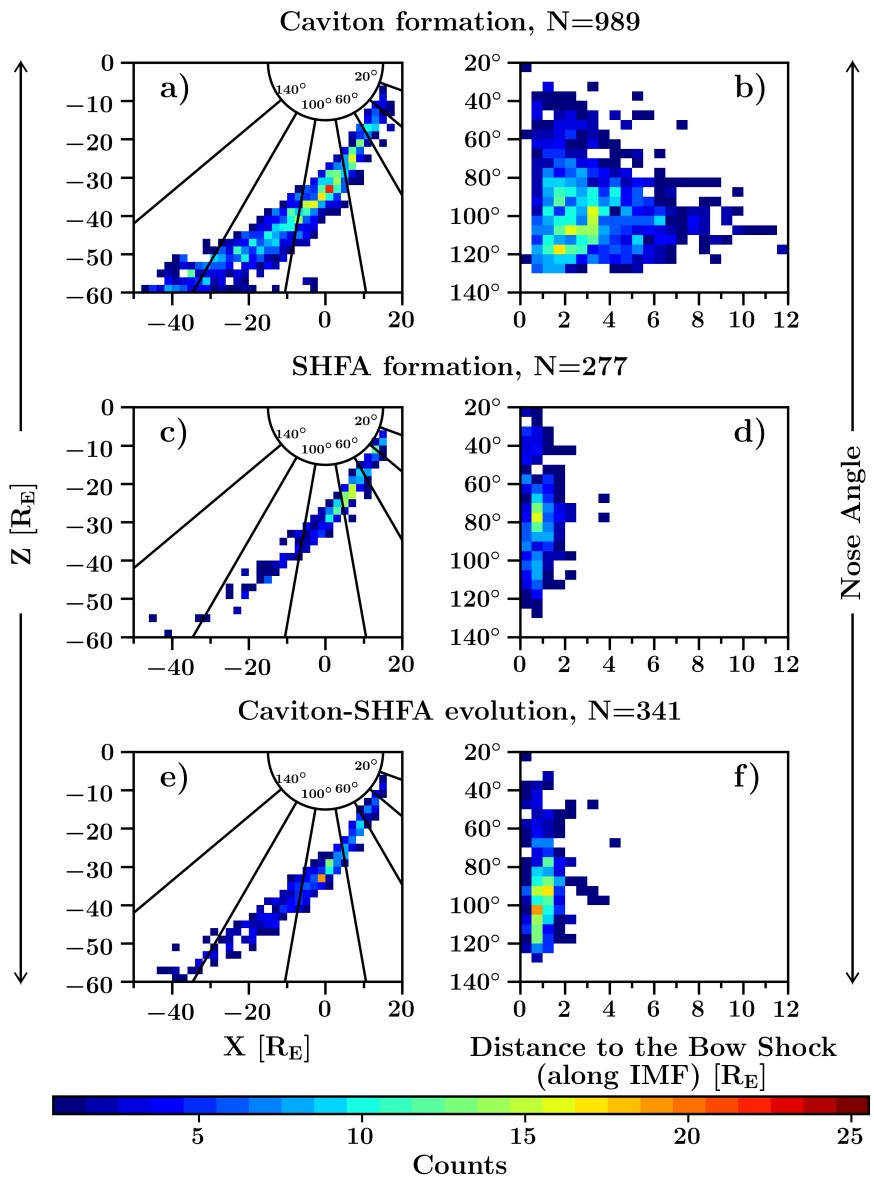

**Figure 5.** 2D distributions of the caviton formation locations (top row), SHFA formation locations (middle row) and the locations where caviton-to-SHFA evolution takes place (bottom row). Panels a), c), e) show the locations in Cartesian GSE-coordinates. Panels b), d), f) show the locations in a distance to the bow shock vs. "nose angle"-coordinate system (see text). An overlay of the nose angle is displayed in panels a), c), e) to ease the comparison between the coordinate systems.



**Figure 6.** Box-and-whiskers plots showing the variation of the tracked transients' physical properties with the distance from the bow shock (along the IMF direction). Each box covers an interval of 1 $R_E$, showing the interquartile data range that contains $50\%$ of the data around the median, which is shown as a black horizontal line. The whiskers extend to $1\%/99\%$ of the data, with outliers displayed as horizontal lines.



**Figure 7.** Box-and-whisker plots showing the variation of the tracked transients' physical properties with the nose angle. Each box covers an angle of $10°$. See Figure 6 for a detailed explanation of the box-and-whiskers plot.

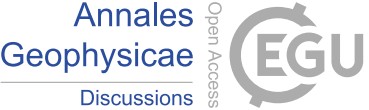

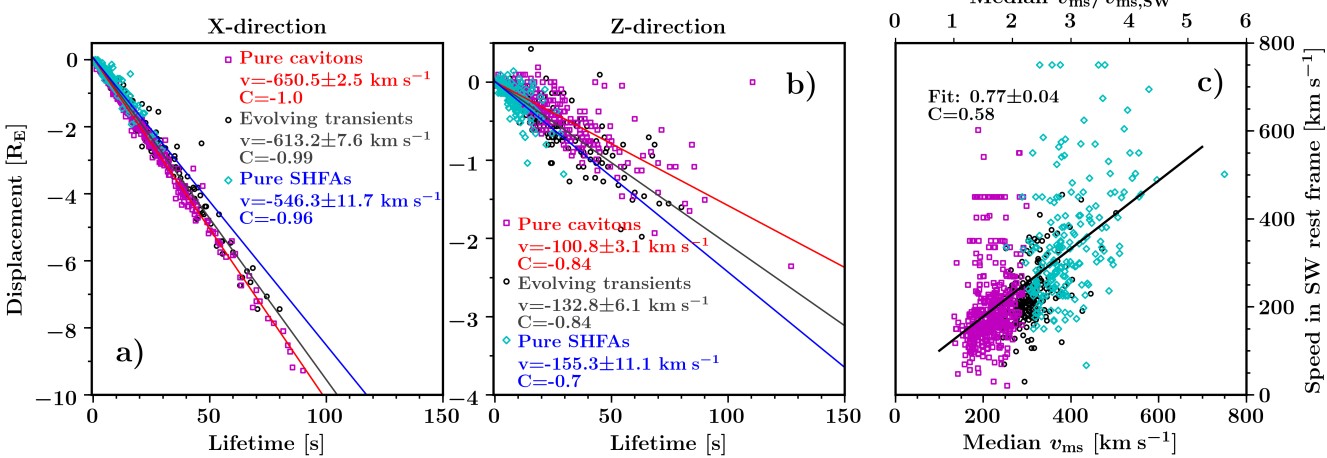

**Figure 8.** Panels a) and b): Scatterplots of the lifetimes of the transients against their displacements in the X-direction and the Z-direction. Linear fits in each panel estimate the propagation velocities for different transient types, including the standard error of the slope of the fit and the Pearson correlation coefficient $C$. Panel c): Scatterplot of the over-lifetime median magnetosonic speeds $v_{ms}$ inside the transients against the solar wind rest frame propagation speeds of the transients.