# Peer review of "Foreshock cavitons and spontaneous hot flow anomalies: A statistical study with a global hybrid-Vlasov simulation"

_Annales Geophysicae, 2020_

## Referee Comment (RC1)

Comments for "Foreshock cavitons and spontaneous hot flow anomalies: A statistical study with a global hybrid-Vlasov simulation" by Tarvus et al. (Paper # angeo-2020-87)

This manuscript presents results of a statistical study of foreshock cavitons and spontaneous hot flow anomalies using a global hybrid-Vlasov simulation. Their properties and evolution have been investigated with interesting results. The manuscript is well written in general and I enjoyed reading it. However, some conclusions are not convincing and the event selection criteria could be improved. My specific comments for improving the manuscript are listed below.

Major:
1. Lines 131-134: About the event selection criteria, I have the following questions:
1a. Since cavitons are embedded in ULF waves in the foreshock, it is crucial to distinguish cavitons from ULF waves. Cavitons should have lower B and N than ambient waves. Is 20% depletion a good enough criterion to distinguish cavitons from ULF waves? What if the background ULF wave amplitude is greater than 20% of Bsw and Nsw? In line 147, it was mentioned that "several transients exhibit elongated shapes and are found together in "chains" that are aligned with the direction of the IMF." Could they be an ULF wave train with amplitude greater than 20% of Bsw and Nsw?

1b. Plasma beta > 10 is used as a criterion to identify SHFAs. Would it be better to use the ion temperature and bulk flow instead of plasma beta to identify SHFAs? First, it is possible that some events with beta > 10 do not show ion heating at all and the high beta is simply due to very low Bt. Using solar wind beta =0.7, for an event with beta = 10 and without heating, B = sqrt(0.07) Bsw =0.26 Bsw. Figure 2b shows that some SHFAs have B below this value. Second, some SHFAs with weak B depletion and moderate heating can also be misidentified as foreshock cavitons. Figure 2c shows that a few SHFAs have low foreshock ion density ratio and some foreshock cavitons have large ratio. Is it possible that these special events were misidentified due to the two reasons mentioned above? Third, the average value of the bulk flow speed inside SHFAs is 19.4% decrease from the solar wind bulk flow speed (lines 210-211). Does this mean that about half of the SHFAs have less than 20% flow decrease and should not be called SHFAs since there is no significant "flow anomaly"?

2. The authors did not compare the plasma properties inside cavitons and SHFAs with the ambient foreshock. Without this comparison, the following conclusions are either not convincing or lack of a physical explanation.

2a. Lines 295-296: "The low amount of suprathermals beyond 1 RE suggests that the accumulation of suprathermals occurs principally very close to the bow shock."
Line 427: "Our results indicate that the accumulation of suprathermals inside cavitons/SHFAs is closely tied to the transients' distance from the bow shock."
Background foreshock ion density, temperature, and velocity need to be used to compare with transient values (increase/decrease ratio). Those background values are also very sensitive to thetaBn and distance from the bow shock. Is there really foreshock ion density increase compared to the background foreshock? The high density ratio could also be due to a decrease in Nsw. Therefore, this may not be due to more accumulation close to the bow shock.

2b. Line 316: "we observe a clear nose angle dependence in the proton temperature"
Line 441: "the temperature inside SHFAs increases towards the bow shock nose."

Is it because near the nose, as foreshock ions are more radially sunward, there are larger relative motion between foreshock ions and solar wind ions causing larger measured ion temperature? I suggest to check the background foreshock ion temperature.

2c. Lines 373-374: "True to their name, the SHFAs in our simulation run are associated with high temperatures and high levels of bulk flow deflection due to the large quantities of suprathermals inside them"
It would be more convincing to say "high temperatures" if they were compared with those in the background foreshock. "high levels of bulk flow deflection" may not be accurate since the average value of the bulk flow speed inside SHFAs is 19.4% decrease from the solar wind bulk flow speed (lines 210-211). See comment 1b above.

Minor:
3. Line 70: "Cavitons were found preferentially during stronger IMF, lower solar wind density and larger solar wind and Alfvén speeds." Is this conclusion based on Figure 16 in Kajdic et al. (2017)? If so, this conclusion may not be correct since the distributions in this figure are not normalized by the background solar wind distributions. If not, please provide the reference.

4. Line 74: This conclusion is based on observations of "19 SHFAs found in the Cluster data between the years 2003 and 2011" by Kajdic et al. (2017). It is very likely that very strict criteria were used and only very significant SHFAs were included in this study because the following studies based on 300 SHFAs from Cluster data and 66 SHFAs from 3 years of THEMIS data showed less than 90% depletion in many SHFAs. Please see Figure 3 in Wang et al. (2013) and Figure 5a in Chu et al. (2017).

Wang, S., Q.-G. Zong, and H. Zhang (2013), Hot flow anomaly formation and evolution: Cluster observations, J. Geophys. Res. Space Physics, 118, 4360–4380, doi:10.1002/jgra.50424.

Chu, C. S., H. Zhang, D. G. Sibeck, Otto, A., Zong, Q., Omidi, N., McFadden, J. P., Fruehauff, D., and Angelopoulos, V. (2017), THEMIS satellite observations of hot flow anomalies at Earth's bow shock, Ann. Geophys., 35, 443-451, doi:10.5194/angeo-35-443-2017.

5. Lines 156, 218-219: Is there any reason to set the lower limit of the event size to 5 cells (0.011 RE)? Why are the transients in the simulation smaller than those observed?

6. Lines 182 and 187: SHFAs tend to be more depleted (up to 94%) than cavitons. This is partially due to the SHFA selection criterion of beta > 10. See comment 1.2 above.

7. Lines 192-193: "Figure 2c shows that nst rarely exceeds 15% of the solar wind density inside cavitons," This is true, but many of them can still have density ratio larger than that in the background foreshock causing higher ion temperature. The ion temperature inside foreshock cavitons should be similar to that that in the ambient foreshock.

8. Lines 203-205: Figure 2d shows that the proton temperature separating cavitons and SHFAs is 14 times the solar wind ion temperature. What is the ion temperature in the background foreshock? Are the ion temperature inside foreshock cavitons similar to that that in the ambient foreshock?

9. Line 213: Could the few examples of cavitons with less than 600 km/s flow speed be SHFAs?

10. Lines 219-220: "Cavitons have a slightly larger average maximum area than SHFAs which could be due to SHFAs forming only near the bow shock, where they do not have time to grow large." This might be true for SHFAs that form independently. How about SHFAs that evolve from cavitons? Shouldn't they be larger than cavitons?

11. Line 271: Should "along the bow shock" be "along the bow shock surface"?

12. Lines 270, 299, 316, 438: The parameters are organized as a function of the nose angle. "There is no single trend controlling the properties of cavitons and SHFAs as the nose angle varies." How about organizing them as a function of thetaBn? "The amount of SHFAs decreases towards the flank of the bow shock." The physics behind this is likely the occurrence rate of SHFAs depends on the thetaBn and the local shock Mach number which decreases towards the flank.

13. Line 371: "we observe a clear difference in the amount of suprathermal protons inside cavitons and SHFAs" There is "a clear difference", but there is also some overlap. What about the ratio of suprathermal protons to Nsw in the ambient foreshock?

14. Line 392: "pick up even the smallest transients that may not be resolvable from spacecraft data amidst ULF waves." Are they really transients or waves? Why are they not resolvable from spacecraft data?

15. Line 446: "larger reductions in the bulk flow speed inside SHFAs near the bow shock nose"
As backstreaming foreshock ions are more sunward, which can reduce more bulk speed (same reason as the high ion temperature near the nose).

---

## Author Response (AR1)

We have revised the manuscript as per the referees' comments, and added references to the changes done in the manuscript at each individual comment below the original responses. The responses to each referee can be found below individually. The revisions in the manuscript are written with a bold font. A general list of all the major changes made in the manuscript can be found at the end of this response letter.
* * *
We thank the referees for providing insightful comments, and believe these will improve the manuscript. Below are responses to each comment individually.

First on a general note, in order to understand the foreshock conditions surrounding the transients, we have compared the plasma properties of the transients to those of the surrounding ULF wave field. We do this by finding the troughs in the ULF wave field, since they can be directly compared to the transients consisting of decreases in density/magnetic field magnitude. This method is illustrated in Figure 1 below. We define the troughs as local minima in the proton number density below the input solar wind value. Unlike cavitons and SHFAs, we do not track the motion of the troughs, but use them only to calculate statistics of various plasma properties (e.g., density, temperature and bulk speed), which are compared to the tracked transients. Only troughs in the relevant region are selected for these statistics (e.g., within 1/4/10 RE from the bow shock). We will refer to these results in the responses below, and also add them to the revised manuscript.

[Figure]

**Above: A plot showing ULF wave troughs / local minima as black dots within 10 RE from the bow shock at time t=900.0 s. The colormap shows values of proton number**

**density below the input solar wind density nSW. The bow shock is modelled with the 4th order polynomial described in the manuscript.**

===== RESPONSES TO REFEREE 1. =====

**Major:**
**1. Lines 131-134: About the event selection criteria, I have the following questions:**

**1a. Since cavitions are embedded in ULF waves in the foreshock, it is crucial to distinguish cavitions from ULF waves. Cavitons should have lower B and N than ambient waves. Is 20% depletion a good enough criterion to distinguish cavitions from ULF waves? What if the background ULF wave amplitude is greater than 20% of Bsw and Nsw? In line 147, it was mentioned that "several transients exhibit elongated shapes and are found together in "chains" that are aligned with the direction of the IMF." Could they be an ULF wave train with amplitude greater than 20% of Bsw and Nsw?**

The choice of a 20% limit is the same as in earlier spacecraft studies by Kajdic et al. (2013, 2017). However, in these studies, the events had to fulfill a subsequent criterion based on a function defined as $\chi(t) = (n(t) - \langle n \rangle) * (B(t) - \langle B \rangle)$ (where n(t) and B(t) are the density and magnetic field magnitude at time t and $\langle \rangle$ denotes a time average). The criterion requires that the value of $\chi$ inside cavitons must be at least 5 standard deviations larger than the temporal average of $\chi$ over the observation interval. We have omitted this subsequent criterion in order to be able to detect small transients and study the temporal evolution of the transients.

   In general, the density and magnetic field magnitude fluctuate ~5-10% from their solar wind values in the foreshock, and the amplitude of the fluctuations is below our caviton detection criteria. Below shown are temporal averages of proton number density np and magnetic field magnitude B over a 120 s interval.

[Figure]

**Above: Temporal averages of proton number density (left) and magnetic field magnitude (right) in the foreshock over a 120 s period.**

More specifically, the amplitude of the general ULF wave field can also be compared to the depth of the transients by considering the depletions at the wave troughs (i.e., local minima) below the input solar wind values. In the region where cavitons are found (< ~10 RE from the bow shock), the proton number density in a trough has a mean depletion of ~12%. Structures with density depletions of >20% represent ~17.4% of all minima in the wave troughs. Based on visual inspection, we consider the 20% limit to be representative of a forming, localised transient. We will add these results concerning the general wave field in the revised manuscript to better motivate the chosen selection criteria.

Chains of cavitons form near the bow shock where the foreshock is permeated by multitude of waves propagating at different angles. These conditions are suitable for repeated formation of cavitons as the cross-propagating ULF waves interact with each other. Visually, such chains are a common feature near the bow shock throughout the simulation. Thus, they likely do not correspond to isolated ULF wave trains, but rather to continuous transient formation.

UPDATE: We have added the comparison on the variation of n and B between foreshock transients and the general ULF wave field in Section 3.1. The remark conserning chains of transients has been added to lines 157-159.

**1b. Plasma beta > 10 is used as a criterion to identify SHFAs. Would it be better to use the ion temperature and bulk flow instead of plasma beta to identify SHFAs? First, it is possible that some events with beta > 10 do not show ion heating at all and the high beta is simply due to very low Bt. Using solar wind beta =0.7, for an event with beta = 10 and without heating, B = sqrt(0.07) Bsw =0.26 Bsw. Figure 2b shows that some SHFAs have B below this value. Second, some SHFAs with weak B depletion and moderate heating can also be misidentified as foreshock cavitons. Figure 2c shows that a few SHFAs have low foreshock ion density ratio and some foreshock cavitons have large ratio. Is it possible that these special events were misidentified due to the two reasons mentioned above? Third, the average value of the bulk flow speed inside SHFAs is 19.4% decrease from the solar wind bulk flow speed (lines 210-211). Does this mean that about half of the SHFAs have less than 20% flow decrease and should not be called SHFAs since there is no significant "flow anomaly"?**

Beta was originally chosen as a criterion for SHFAs due to the large variation found in the temperature throughout the foreshock, which makes setting an explicit temperature criterion challenging. Similarly, we did not wish to make assumptions on the bulk flow speeds inside the transients. The physical motivation behind the choice of beta is that a large beta indicates that the interiors of the structures are dominated by the plasma instead of the magnetic field. The value of beta > 10 was chosen based on visual inspection. In the end, we retained the beta critetion as it appears to pick the differences between cavitons and SHFAs well, and in order to keep our results comparable with the earlier Vlasiator caviton/SHFA study by Blanco-Cano et al. (2018), where the beta-criterion was originally used. We acknowledge that other SHFA criteria could be also succesfully applied. If a temperature criterion was used instead, a similar classification to the present would be obtained, as panel d) in Figure 2 shows.

On the possibility of transients being miscategorised due to low magnetic field magnitude, we note that all SHFAs in panel d) of Figure 2 have temperatures above the mean temperature of cavitons, and conclude that the impact of the magnetic field

magnitude on the categorisation is not statistically significant. For the specific case of transients with B depletions below 0.26 Bsw, we find that the temperature inside these transients ranges between ~8-40 MK, and conclude that they are correctly categorised as SHFAs. For the suprathermal density, we see in panels b) and c) of Figure 3 that for each temperature/bulk flow speed, a large range of suprathermal densities is found, and overlap between cavitons and SHFAs exists although they have different temperatures and bulk flow speeds.

    Finally, it is true that there is a large proportion of SHFAs with minor flow deflections. These might be better described as "proto-SHFAs", similar to young structures discussed in e.g., Zhang et al. (2013). We will discuss this in the revised manuscript.

UPDATE: We have added brief motivation on the beta criterion on lines 141-144 in the Methods-section, and discuss the validity of the criterion in the Discussion section on lines 416-433. We also discuss the nature of the SHFAs with weak bulk flow deflections (i.e., "proto-SHFAs") on lines 431-433 and lines 543-546.

**2. The authors did not compare the plasma properties inside cavitons and SHFAs with the ambient foreshock. Without this comparison, the following conclusions are either not convincing or lack of a physical explanation.**

Since the conditions in the foreshock vary both in space and time, defining ambient properties in the foreshock is difficult. We have evaluated these conditions by considering the troughs in the foreshock ULF wave field, which can be directly compared to the foreshock transients. We define the troughs as local minima of proton number density below the input solar wind value. We will supplement the manuscript to include discussion on this comparison. See below for each individual point.

UPDATE: We have included this comparison in Section 3.1.

**2a. Lines 295-296: "The low amount of suprathermals beyond 1 RE suggests that the accumulation of suprathermals occurs principally very close to the bow shock." Line 427: "Our results indicate that the accumulation of suprathermals inside cavitons/SHFAs is closely tied to the transients' distance from the bow shock." Background foreshock ion density, temperature, and velocity need to be used to compare with transient values (increase/decrease ratio). Those background values are also very sensitive to thetaBn and distance from the bow shock. Is there really foreshock ion density increase compared to the background foreshock? The high density ratio could also be due to a decrease in Nsw. Therefore, this may not be due to more accumulation close to the bow shock.**

As we compare the number density of suprathermal ions directly to the input solar wind ion number density, the presented values of the suprathermal ion density represent an absolute increase in the amount of suprathermals. Within 1 RE from the bow shock, the mean suprathermal ion density in a trough of a ULF wave is ~0.01 nSW. Panel h) in Figure 6 shows that within 1 RE from the bow shock, the number density of suprathermal ions inside SHFAs (~0.05-0.78 nSW) typically exceeds this value. The high amounts of suprathermals inside SHFAs are associated with the rippled bow shock surface, and are not uniformly present in the foreshock.

UPDATE: The above response contains a typo in the value ~0.01 nSW, where it should read ~0.1 nSW. We have added the comparison of the suprathermal number density inside foreshock transients and in the general wave field in Section 3.1, lines 221-225.

**2b. Line 316: "we observe a clear nose angle dependence in the proton temperature" Line 441: "the temperature inside SHFAs increases towards the bow shock nose." Is it because near the nose, as foreshock ions are more radially sunward, there are larger relative motion between foreshock ions and solar wind ions causing larger measured ion temperature? I suggest to check the background foreshock ion temperature.**

As the temperature is evaluated in the plasma bulk frame, both the sunward and lateral components of the velocity affect the calculated temperature of suprathermal ions. In general, the suprathermal population behaves as a quasi-gyrating population that travels along the mean magnetic field direction. The largest field-parallel velocities are found near the foreshock edge, and the velocity decreases deeper into the foreshock, explaining the higher temperatures and lower bulk flow speeds near the shock nose. We will add discussion on this effect to the revised manuscript.

UPDATE: We have included this point on lines 358-360 and 504-506 in the manuscript.

**2c. Lines 373-374: "True to their name, the SHFAs in our simulation run are associated with high temperatures and high levels of bulk flow deflection due to the large quantities of suprathermals inside them" It would be more convincing to say "high temperatures" if they were compared with those in the background foreshock. "high levels of bulk flow deflection" may not be accurate since the average value of the bulk flow speed inside SHFAs is 19.4% decrease from the solar wind bulk flow speed (lines 210-211). See comment 1b above.**

In the region where SHFAs are found (within ~4 RE from the bow shock), the mean temperature in a trough of a ULF wave is ~4.3 MK, a bit over half the approximate minimum temperature inside SHFAs (>~7 MK). While the mean decrease in the bulk flow speed inside SHFAs (19.4%) is relatively small, it is roughly 4 times larger than the mean bulk flow speed decrease inside ULF wave troughs (4.8%) in the region where SHFAs are present. However, "high levels of bulk flow deflection" might be better described as "higher levels of bulk flow deflection than cavitons". We will revise this expression.

UPDATE: Comparison between the temperatures and the bulk flow deflections inside the transients and in the general foreshock wave field has been included in Section 3.1 on lines 228-247.

**Minor:**
**3. Line 70: "Cavitons were found preferentially during stronger IMF, lower solar wind density and larger solar wind and Alfvén speeds." Is this conclusion based on Figure 16 in Kajdic et al. (2017)? If so, this conclusion may not be correct since the distributions in this figure are not normalized by the background solar wind distributions. If not, please provide the reference.**

The conclusion is taken from section 2.2.1 in Kajdic et al. (2013), where the background values are taken into consideration. Admittedly, the phrasing is ambiguous on line 70. This will be rephrased in the manuscript as: "Kajdic et al. (2013) observed cavitons preferentially during stronger IMF, ..."

UPDATE: This has been fixed on line 70 in the manuscript.

**4. Line 74: This conclusion is based on observations of "19 SHFAs found in the Cluster data between the years 2003 and 2011" by Kajdic et al. (2017). It is very likely that very strict criteria were used and only very significant SHFAs were included in this study because the following studies based on 300 SHFAs from Cluster data and 66 SHFAs from 3 years of THEMIS data showed less than 90% depletion in many SHFAs. Please see Figure 3 in Wang et al. (2013) and Figure 5a in Chu et al. (2017).**

The statement on line 74 will be supplemented with the provided references as follows:

"...However, the magnitude of the depletions inside SHFAs listed by Kajdic et al. (2017) may be a product of the strict criteria used to detect the events. Other spacecraft observations of potential SHFAs by Wang et al. (2013) and Chu et al. (2017) show numerous examples with depletions having magnitudes less than 90%."

UPDATE: We have added these references on lines 75-78.

**5. Lines 156, 218-219: Is there any reason to set the lower limit of the event size to 5 cells (0.011 RE)? Why are the transients in the simulation smaller than those observed?**

A lower size limit is employed due to our automated transient tracking method. Since the method is based on the overlap between transients at consequent timesteps of the simulation, a minimum size limit ensures that small transients can be tracked consistently. A minimum size of 5 cells was selected as a good trade-off to ensure consistent tracking and a large sample of transients.

  The sizes of the transients in our simulations are affected by the following factors; First, the spatial resolution of the simulation can limit the steepening of ULF waves, also limiting the sizes of foreshock transients, as discussed also in Blanco-Cano et al (2018). Second, as we define the transients as structures below 80% of solar wind ion density/magnetic field magnitude, only the area below this limit is taken into account. This does not take into account the "shoulders" surrounding the transients, where the density/magnetic field magnitude are enhanced. These shoulders typically have a finite width, and they are not identified by our automated transient detection algorithm. Finally, the size difference can partially result from the selection of the transients. Our data set includes a number of small transients which might be discarded by selection criteria used in past spacecraft studies.

UPDATE: We have added motivation on the lower size limit on line 168. A note on the effect of the sample selection on the transient size has been added on lines 446-449.

**6. Lines 182 and 187: SHFAs tend to be more depleted (up to 94%) than cavitons. This is partially due to the SHFA selection criterion of beta > 10. See comment 1.2 above.**

For the large majority of SHFAs in our study, the relative increase in the temperature (T) is much greater than the relative decrease in the magnetic field strength (B). While beta has a stronger dependence on B than T, the decrease in B is countered to some extent by the comparable decrease in plasma number density (n), so that beta = $nT/B^2$ behaves as ~T/B. Since the change in T is much larger than the change in B, B should not impact the classification of the transients.

UPDATE: As indicated in response 1b, we have added discussion on the validity of the beta-criterion in the Discussion-section on lines 416-433.

**7. Lines 192-193: "Figure 2c shows that nst rarely exceeds 15% of the solar wind density inside cavitons," This is true, but many of them can still have density ratio larger than that in the background foreshock causing higher ion temperature. The ion temperature inside foreshock cavitons should be similar to that in the ambient foreshock.**

The temperature in the foreshock is dependent on the distance to the bow shock, and a similar temperature dependence is found inside both cavitons and the surrounding ULF waves. For all troughs within 10 RE from the shock, the mean temperature is ~2.8 MK. Panel i) in Figure 6 shows that the temperatures inside cavitons far from the shock (~5-10 RE) are comparable to this value. Near the shock (< ~2 RE), where the temperatures inside cavitons are considerably higher, there is a similar temperature increase inside ULF wave troughs, showing a mean temperature of ~5.8 MK within that range. We will add discussion about the temperature to the manuscript.

UPDATE: Comparison of the suprathemal density inside transients and in the general foreshock wave field has been added in Section 3.1 on lines 221-225, as indicated above in response 2a.

**8. Lines 203-205: Figure 2d shows that the proton temperature separating cavitons and SHFAs is 14 times the solar wind ion temperature. What is the ion temperature in the background foreshock? Are the ion temperature inside foreshock cavitons similar to that in the ambient foreshock?**

Please see the response no. 7 above. The foreshock contains a range of temperatures, and the temperature generally depends on the location in the foreshock. Hence, it is not possible to define a single foreshock temperature. The temperatures inside cavitons and ULF waves are similar, and show similar dependence on location, as demonstrated above.

UPDATE: Comparison between the temperatures inside the transients and in the general foreshock wave field has been included in Section 3.1 on lines 228-236.

**9. Line 213: Could the few examples of cavitons with less than 600 km/s flow speed be SHFAs?**

The majority of the cavitons having such low bulk flow speeds are evolving transients, which fulfill our SHFA classification later in the simulation (7 out of 11 transients). In these

cases, the overall minimum of bulk flow speed has occurred while the transients were still classified as cavitons.

The last four cavitons that do not evolve into SHFAs according to our criteria, propagate close to the bow shock (≤ 1 RE). Their temperatures are in the range 5-7 MK. According to panels i) and j) in Figure 6, these temperatures are in the low end of SHFA temperatures at those distances. These examples may be cavitons that are beginning to evolve into SHFAs, but do not reach a fully developed stage before they disappear near the shock.

UPDATE: We have included the remark on cavitons with low bulk speeds evolving into SHFAs beginning on line 243. We also discuss the lack of simultaneous heating/bulk flow deflection on lines 245-247, 430-433 and 543-546.

**10. Lines 219-220: "Cavitons have a slightly larger average maximum area than SHFAs which could be due to SHFAs forming only near the bow shock, where they do not have time to grow large." This might be true for SHFAs that form independently. How about SHFAs that evolve from cavitons? Shouldn't they be larger than cavitons?**

The statement on lines 219-220 does not account for the evolution of the transients, so that independently forming SHFAs and SHFAs evolving from cavitons are counted in the same category. When they are considered separately, it is indeed seen that cavitons evolving into SHFAs have the largest average size, followed by cavitons that do not evolve and SHFAs that do not evolve from cavitons:
- Cavitons evolving into SHFAs;      ~0.27 RE
- Cavitons not evolving into SHFAs: ~0.11 RE
- Independently forming SHFAs:      ~0.08 RE

This will be clarified in the revised manuscript.

UPDATE: This has been added on lines 253-255.

**11. Line 271: Should "along the bow shock" be "along the bow shock surface"?**

This will be rephrased as suggested.

UPDATE: This has been implemented.

**12. Lines 270, 299, 316, 438: The parameters are organized as a function of the nose angle. "There is no single trend controlling the properties of cavitons and SHFAs as the nose angle varies." How about organizing them as a function of thetaBn? "The amount of SHFAs decreases towards the flank of the bow shock." The physics behind this is likely the occurrence rate of SHFAs depends on the thetaBn and the local shock Mach number which decreases towards the flank.**

ThetaBn is not used in this study since most of the analysed transients are located at the flank of the bow shock, where thetaBn has a narrow value range. Thus, we chose to use the nose angle instead, as it is better suited for analysing the spatial variation of transient formation and properties. We will add discussion on the range of thetaBn in the revised manuscript.

UPDATE: We have added motivation on the choice of the nose angle on lines 310-311 and discuss the effect of thetaBn on transient formation on lines 318-322.

**13. Line 371: "we observe a clear difference in the amount of suprathermal protons inside cavitons and SHFAs" There is "a clear difference", but there is also some overlap. What about the ratio of suprathermal protons to Nsw in the ambient foreshock?**

A general estimate of the suprathermal proton number density in the foreshock surrounding the transients can be obtained by looking at the number densities in the troughs of ULF waves. In the region where both cavitons and SHFAs are present (within ~4 RE from the bow shock), the suprathermal proton density at a trough of a ULF wave has a mean value of ~0.05 nSW. This value is similar to that found inside cavitons, and lower than the values found inside SHFAs.

UPDATE: As stated in response 2a, we have added comparison on the suprathermal number density inside foreshock transients and the general wave field in Section 3.1, lines 221-225.

**14. Line 392: "pick up even the smallest transients that may not be resolvable from spacecraft data amidst ULF waves." Are they really transients or waves? Why are they not resolvable from spacecraft data?**

Since cavitons evolve from interacting ULF waves, there is no clear threshold for identifying an event in which a caviton forms from the ULF wave field. Due to our automated detection method, our study includes small structures that might be discarded by the more stringent selection criteria used in spacecraft observations, such as those employed by Kajdic et al. 2013. In spacecraft data, only fully developed transients can be detected, i.e., they need to be visually identifiable from the surrounding ULF waves. We will rephrase the sentence quoted in the comment as follows to make its meaning clearer:

"...pick up even the smalles transients that may not be identifiable from spacecraft data amidst ULF waves."

UPDATE: This sentence has been rephrased beginning on line 446.

**15. Line 446: "larger reductions in the bulk flow speed inside SHFAs near the bow shock nose" As backstreaming foreshock ions are more sunward, which can reduce more bulk speed (same reason as the high ion temperature near the nose).**

We will include this point in the revised manuscript.

UPDATE: This point has been added on line 363.

**===== RESPONSES TO REFEREE 2. =====**

**MAJOR:**

**Throughout there is very little explicit comparison of the properties of the transients compared to the foreshock in general, let alone the ambient foreshock at the transient's location. Instead mostly only values in the pristine solar wind are used for comparison. However, understanding how the structures differ from their surroundings is of vial importance and needs to be incorporated into the work throughout. This affects numerous aspects of the work, including:**

**\* Are the choices of properties and thresholds for detection of the transients suitable? How does a 20% decrease in density compare to the variability in density associated with the foreshock ULF wave field? Is plasma beta a sensible parameter to use to distinguish between cavitons and SHFAs (I would have thought a temperature criterion would have been more appropriate) and how does a value of 10 compare to the typical foreshock and its variability?**

The choice of a 20% limit is the same as in earlier spacecraft studies by Kajdic et al. (2013, 2017). However, in these studies, the events had to fulfill a subsequent criterion based on a function defined as $\chi(t) = (n(t) - <n>) * (B(t) - )$ (where $n(t)$ and $B(t)$ are the density and magnetic field magnitude at time t and $<>$ denotes a time average). The criterion requires that the value of $\chi$ inside cavitons must be at least 5 standard deviations larger than the temporal average of $\chi$ over the observation interval. We have omitted this subsequent criterion in order to be able to detect small transients and study the temporal evolution of the transients.

    In general, the density and magnetic field magnitude fluctuate ~5-10% from their solar wind values in the foreshock, and the amplitude of the fluctuations is below our caviton detection criteria. This is demonstrated below, where the temporal averages of proton number density np and magnetic field magnitude B are shown over a 120 s interval.

[Figure]

**Above: Temporal averages of proton number density (left) and magnetic field magnitude (right) in the foreshock over a 120 s period.**

More specifically, the depths of the transients can also be compared to those of the surrounding ULF waves by taking into account each trough (i.e., local minima) in the foreshock below the input solar wind density. In the region in which cavitons are present (< ~10 RE from the bow shock), the mean depth of a trough in the wave field is ~12%, a bit over half of the caviton detection criterion. Structures below the 20% limit represent ~17.4% of all troughs in this region. We will add these results concerning the general wave field in the revised manuscript to better motivate the chosen detection criteria.

Beta was chosen as the SHFA criterion due to the large variation of temperature in the foreshock, which makes choosing an explicit temperature condition challenging. The physical motivation behind the choice of beta is that a large beta indicates that the transients are dominated by the plasma instead of the magnetic field. In the end, we retained the beta criterion as it appears to pick the differences between cavitons and SHFAs well, and in order to keep our results comparable with the earlier Vlasiator caviton/SHFA study by Blanco-Cano et al. (2018), where the beta-criterion was originally used. A value of 10 was chosen visually. In the region where cavitons are found (< ~10 RE), the beta in a ULF wave trough has a mean value of ~4.5, well below our criterion. Values below 10 represent 92.3% of all troughs. Finally, we acknowledge that other SHFA criteria could be also applied. If a temperature criterion was used instead, we would obtain a similar classification, as panel d) in Figure 2 shows.

UPDATE: We have added the comparison on the variation of n and B between foreshock transients and the general ULF wave field in Section 3.1. We have also added a brief motivation on the beta-criterion on lines 141-144, and discussion on its validity on lines 416-433 in the Discussion section.

**\* In Figure 2, how do the density of suprathermals and temperatures of cavitons and SHFAs compare to typical foreshock conditions? Are the velocities in these structures significantly different from the ambient?**

Using the same method as above, the suprathermal densities and temperatures inside cavitons/SHFAs and ULF waves can be compared by considering the values at the troughs in the wave field. However, both the suprathermal density and the temperature are sensitive to the location in the foreshock, with both increasing rapidly near the shock. We will supplement the revised manuscript with the below results.

In the region where most cavitons are found (< ~10 RE from the shock), the suprathermal density in a trough of a wave has a mean value of ~0.03 nSW. Panel g) in Figure 6 shows that the suprathermal density inside cavitons far from the shock is similar to this value. Near the shock, where both cavitons and SHFAs are present (< ~4 RE), the mean suprathermal density in a wave trough is ~0.05 nSW, and a comparable increase is found inside cavitons.

For the temperature, the corresponding mean values at <10 RE and <4 RE are 2.8 MK and 4.3 MK, respectively. Comparing these values to panel i) of Figure 6 shows that the temperature inside cavitons is similar for both ranges. As stated in the manuscript, the suprathermal densities and temperatures are larger inside SHFAs than inside cavitons, and thus by extension, larger compared to the surrounding ULF wave field.

Since the bulk velocity is also dependent on the suprathermal density, it shows a general decrease towards the bow shock. Within (10, 4, 1) RE from the shock, a mean bulk speed of (~728 km/s, ~714 km/s, ~680.2 km/s) is found in a wave trough. The speeds inside cavitons are similar to these values.

UPDATE: We have supplemented the manuscript with discussion on the suprathermal density (lines 216-225), temperature (lines 228-236) and bulk flow speed (lines 237-247), comparing the transients and the general foreshock wave field.

**\* In Figure 3, are the correlations presented simply extensions of the overall foreshock or do they constitute distinct populations?**

For the proton number density and magnetic field magnitude, there exists a continuity across fluctuations within 20% of the ambient solar wind values to cavitons/SHFAs. Since the transients form from the ULF wave field, there is no clear limit between a ULF wave and a transient. However, the density and magnetic field magnitude are relatively well correlated inside structures surpassing the 20% limit, as shown in panel a) of Figure 3. When all troughs are taken into account, it is seen that the values of the density and magnetic field magnitude appear spread for small-magnitude fluctuations. For larger depressions, the parameters become correlated. This is illustrated in the figure attached below.
   For the suprathermal density, temperature and bulk flow speed, the distributions for density fluctuations within 20% of the solar wind density exhibit similar shapes as shown in panels b) and c) of Figure 3. The values are concentrated near their respective solar wind values, having similar ranges as cavitons. The 90th (10th) percentile values for the suprathermal density and temperature (bulk flow speed) are 0.07 nSW and 5.6 MK (690.6 km/s), respectively.

UPDATE: The below Figure contains an error as only a part of the studied simulation interval was mistakenly included in the data. The full data does not show significant change in the correlation between the foreshock transients and smaller ULF waves. Thus, we conclude that they constitute the same population as the transients develop from the ULF wave field. We have included this observation in the revised manuscript on lines 264-266.

[Figure]

**Above: A scatterplot of proton number density (n) and magnetic field magnitude (B) for all ULF wave troughs below the input solar wind number density.**

**\* I also have concerns over the results surrounding the suprathermal ions. The method employed of distinguishing between core and suprathermals uses the velocity and temperature of the pristine solar wind. This seems unsuitable for transients associated with flow anomalies, as the authors concede on line 200, and thus many of the results are likely micharacterising the solar wind and suprathermal ions in these structures. I would suggest the authors reprocess the data separating out regions in phase space using a distance condition in velocity space (based on the temperature in the pristine solar wind) either from the bulk or peak phase space density.**

Unfortunately, the recategorisation of the core and suprathermal populations is not possible post-run. The suprathermal population is resolved from the velocity distribution function as the simulation is running, and the distribution function is saved afterwards only in selected simulation cells to keep the sizes of the simulation bulk files tractable (of the order of GB as opposed to TB for a file with the velocity distribution available everywhere). Input solar wind values are utilised in the categorisation since the process is automated during the simulation run.

We also note that the method is valid in the majority of the foreshock (excluding major hot flow anomalies), as the deflections seen in SHFAs are a result of the velocity moments, as opposed to deceleration of the solar wind core. This effect is demonstrated for HFAs by Parks et al. (2013).

UPDATE: We have included the above remark on the validity of the method for distinguishing the core/suprathermal populations on lines 211-215.

**\* Related to the above, many conjectures around how the solar wind beam vs. the suprathermals are affecting the moments of the distribution are made, however, no velocity distributions are presented within the manuscript. It is known that the distrbutions within foreshock transients can evolve from multicomponent to single component plasmas, whereas the authors posit only the former.**

Since the manuscript already contains quite a lot of material with several large figures, we feel that the scope of this study should be limited to presenting statistics of the general properties and evolution of foreshock transients. In this regard, we propose to reduce the emphasis on the velocity distributions in the revised manuscript, and leave detailed study of their evolution as a topic for future work. The structure of foreshock velocity distributions is however briefly discussed in the previous Vlasiator study of cavitons/SHFAs by Blanco-Cano et al. (2018), and more extensively in a similar simulation performed by Battarbee et al. (2020), who studied the helium foreshock using Vlasiator.

UPDATE: We have limited the technical discussion on velocity distribution types to our definition of solar wind core and foreshock suprathermal populations on lines 209-215.

**\* Finally, the results with relation to the "nose angle" (which may be better described in the manuscript as meridional angle or solar zenith angle throughout) need to be understood in terms of the theta_Bn angle that the transient is magnetically connected to, since this largely controls the physics of the foreshock. This may aid in the interpretation of the results.**

ThetaBn is not used in this study since most of the analysed transients are located at the flank of the bow shock, where thetaBn has a narrow value range. Thus, we chose to use the nose angle instead, as it is better suited for analysing the spatial variation of transient formation and properties. We will add discussion on the range of thetaBn in the revised manuscript.

UPDATE: This motivation on the choice of nose angle has been added to lines 310-311. We have added discussion on the role of thetaBn on transient formation on lines 318-322.

**MINOR:**

**Lines 20-21: "before it is deflected by the magnetopause" This could do with rewording, since the bow shock also deflects the solar wind and the pressure gradients present throughout the magnetosheath (between bow shock and magnetopause) act to deflect the plasma around the boundary.**

The sentence will be reworded followingly:

"The bow shock slows the solar wind down to submagnetosonic speeds while compressing and heating it. This allows the solar wind to flow around the magnetopause that separates the solar wind from the magnetosphere."

UPDATE: This has been implemented on lines 19-21.

**Line 23: "far back into the upstream." This is not true for the entire region of the shock connected to the IMF, as the sentence suggests, only in the quasi-parallel case. Please reword this sentence, for example, removing the word "far".**

The sentence containing this phrase will be rephrased as follows:

"At the quasi-parallel part of the bow shock (defined as the region where the shock normal and the IMF have an angle theta_Bn < 45), ions reflected off the shock can propagate several hundred ion inertial lengths upstream, forming the foreshock region. At the quasi-perpendicular shock (theta_Bn < 45), the upstream motion of the reflected ions is limited to an order of ion gyroradius, and a more abrupt shock crossing is found."

UPDATE: This has been implemented on lines 21-25.

**Line 59: "SHFAs evolve" I would say they are "thought to evolve" since this is point requires further evidence in general and the results of the manuscript show it be the case only for some SHFAs.**

This will be rephrased as requested.

UPDATE: This has been changed in the abstract and on line 59.

**Line 188: "SHFAs tend to be more depleted than cavitons" This could simply be an effect of the plasma beta condition so needs further comment.**

For the large majority of SHFAs in our study, the relative increase in the temperature (T) is much greater than the relative decrease in the magnetic field strength (B). While beta has a stronger dependence on B than T, the decrease in B is countered to some extent by the comparable decrease in plasma number density (n), so that beta = nT/B^2 behaves as ~T/B. Since the change in T is much larger than the change in B, B should not impact the classification of the transients.

UPDATE: We have added discussion on this observation and the validity of the beta-criterion on lines 416-433.

**Figure 4: PDFs would be more helpful to readers than CDFs to see the regions where the transients actually form, rather than cumulatively from the bow shock up to some region where a certain proportion form. Some of the cumulative numbers can remain in the text, however.**

We will change Figure 4 to the figure attached below, so both PDFs and CDFs are shown.

UPDATE: This has been implemented in the manuscript.

**Table 1: Minimina and maxima of probability distributions are not robust statistics, the 25th and 75th percentile would be more appropriate columns to use. This would also remove potential confusion between the minimum and maximum value for each a particular transient used in the left column, which is appropriate.**

This will be changed.

UPDATE: This has also been implemented in the manuscript.

**Figure 4: The label states these are counts, but they are proportions**

This will be fixed.

UPDATE: This has been fixed in the updated figure.

[Figure]

**Above: Updated Figure 4 displaying both PDFs and CDFs.**

**SUMMARY OF MAJOR REVISIONS:**

**Section 3.1:** Incorporated comparison between the properties of the studied foreshock transients and the surrounding foreshock throughout the section.

**Section 3.2, Figure 4:** Modified Figure 4 to show both the probability density functions and the cumulative probability density functions of the formation locations of cavitons and SHFAs and the locations of caviton-to-SHFA evolution. Accommodated Section 3.2 to match the updated Figure.

**Section 3.2, L310-311, 318-322:** Added motivation on the choice of nose angle over thetaBn and discussion on the role of thetaBn on transient formation.

**Chapter 4, L416-433:** Added validation of the beta-criterion used for classifying the transients.

**L245-247, L428-433, L543-546:** Added discussion on SHFAs with weak bulk flow deflections / cavitons with moderate bulk flow deflections and their implications on the transient evolution.

**Table 1:** Modified the table to show the 25% / 75% percentile values of the presented statistical quantities.

---

## Author Response (AR2)

24.09.2021

We thank the referees and the editor for reviewing the revised manuscript. The final corrected manuscript contains the following corrections as per the referees' comments:

*L143: Added the range of typical values of β in the foreshock surrounding the transients.

*L212: The paragraph concerning the separation of solar wind and suprathermal populations has been corrected to better reflect spacecraft observations inside transients, including references to the papers provided by the referee.

Thank you,
-Vertti Tarvus